# TSM-Bench: Detecting LLM-Generated Text in Real-World Wikipedia Editing Practices

**Gerrit Quaremba**[1]**, Elizabeth Black**[1]**, Denny Vrandečić**[2]**, Elena Simperl**[1]

[1]King's College London, [2]Wikimedia Foundation

`gerrit.quaremba@kcl.ac.uk`

## Abstract

Automatically detecting machine-generated text (MGT) is critical to maintaining the knowledge integrity of user-generated content (UGC) platforms such as Wikipedia. Existing detection benchmarks primarily focus on *generic* text generation tasks (e.g., "Write an article about machine learning."). However, editors frequently employ LLMs for specific writing tasks (e.g., summarisation). These *task-specific* MGT instances tend to resemble human-written text more closely due to their constrained task formulation and contextual conditioning. In this work, we show that a range of SOTA MGT detectors struggle to identify task-specific MGT reflecting real-world editing on Wikipedia. We introduce TSM-Bench, a multilingual, multi-generator, and *multi-task* benchmark for evaluating MGT detectors on common, real-world Wikipedia editing tasks. Our findings demonstrate that (*i*) average detection accuracy drops by 10–40% compared to prior benchmarks, and (*ii*) a generalisation asymmetry exists: fine-tuning on task-specific data enables generalisation to generic data—even across domains—but not vice versa. We demonstrate that models fine-tuned exclusively on generic MGT overfit to superficial artefacts of machine generation. Our results suggest that, in contrast to prior benchmarks, most detectors remain unreliable for automated detection in real-world contexts such as UGC platforms. TSM-Bench therefore provides a critical foundation for developing and evaluating future models.

## 1 Introduction

Wikipedia serves as a vital source of reliable human-written text (HWT) for the artificial intelligence (AI) community. As one of the largest high-quality multilingual corpora on the internet, it features in the training data of most large language models (LLMs) (Deckelmann, 2023; Longpre et al., 2024). However, the Wikimedia Foundation warns that the proliferation of machine-generated text (MGT) across Wikipedia could undermine its knowledge integrity.[1] The unchecked spread of MGT risks degrading the very data underpinning much of recent progress in AI. Generative models trained on uncurated data may deteriorate over time, potentially resulting in fatal model collapse (Shumailov et al., 2024). Therefore, differentiating MGT from HWT is an essential task with wide-ranging downstream implications, resulting in automatic MGT detection becoming an active area of research (Wu et al., 2025a).

Prior work on benchmarking MGT detectors (e.g. Guo et al., 2023; Macko et al., 2023; Li et al., 2024; He et al., 2024; Wang et al., 2024b;a; Wu et al., 2024) has largely relied on simple text generation prompts such as: "*Write an article about machine learning.*" In practice, however, editors typically employ LLMs to support a range of specific writing tasks (Ford et al., 2023; Zhou et al., 2025). Compared with earlier work's free-form **generic** text generation, prompts in real-world editing scenarios are narrower in scope and often contextually constrained (e.g., summarisation). We refer to this as **task-specific** text generation. Crucially, generic and task-specific text differ in that the former is often linguistically and semantically less similar to human text, whereas the latter—because of its constraints—tends to align more closely in style and meaning. Figure 1 illustrates this distinction across four metrics, comparing HWT, generic and the task-specific MGT generated

---

[1]Wikipedia Community Call Notes 2023–24

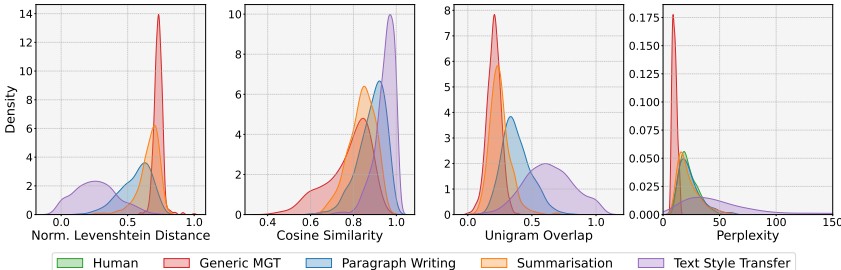

Figure 1: Comparison of textual characteristics between human text, generic MGT, and three task-specific English MGTs. Task-specific MGTs more closely resemble human text. The same pattern is observed in other languages (see Appendix D.2).

for the three Wikipedia editing tasks considered in this study. As detectors learn from such textual patterns, it is well established that detection performance decreases as the total variation distance between human and machine distributions narrows (Sadasivan et al., 2023). We therefore expect detectors to face greater challenges on task-specific data than on prior benchmarks limited to generic data. Assessing their reliability in more realistic MGT scenarios is critical, as detectors safeguard the content integrity of UGC and thus ensure high-quality, uncontaminated data for downstream use across diverse AI applications.

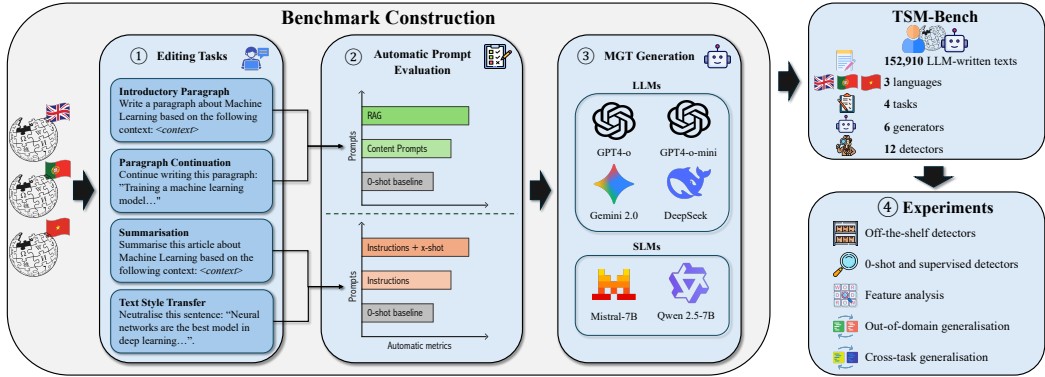

Figure 2: Overview of TSM-BENCH: ① We define four editing tasks informed by research on how editors employ LLMs. ② For each task, we adopt two prompts from the natural language generation literature and automatically evaluate them against a simple baseline. ③ Using the highest-scoring prompt, we generate MGT from six LMs. ④ Finally, we run five experiments on these data and draw key conclusions about the effectiveness of detectors in identifying real-world MGT instances.

In this work, we introduce **T**ask-**S**pecific **M**GT Benchmark (TSM-BENCH), a multilingual, multi-generator, and *multi-task* MGT detection benchmark (see Figure 2), designed to move beyond generic MGT detection and evaluate detectors on text that more closely reflects how users employ LLMs in real-world workflows. Drawing on real-world accounts of how Wikipedia editors use LLMs (Ford et al., 2023; Zhou et al., 2025), we define four common editing tasks. For each task, we adopt two prompts from the natural language generation literature and compare them with a minimal baseline using automatic metrics. We then employ the highest-scoring prompts to generate MGT with six LMs of varying sizes across three languages (English, Portuguese, and Vietnamese). We select languages with different resource levels to study communities beyond the frequently examined English Wikipedia. We define resource level using two indicators: (1) the number of active Wikipedia users and (2) the language's share in the Common Crawl corpus,[2] a standard training source for LLMs.[3] On these data, we conduct extensive experiments benchmarking 12 SOTA de-

---

[2] https://commoncrawl.org/
[3] Appendix Table 5 reports the metrics used to guide our language selection.

tectors from different model families, testing their out-of-domain and cross-task generalisability. Finally, to strengthen our case for moving beyond generic MGT, we analyse feature importance in models trained on generic versus task-specific data.

Our contributions are as follows:

- **Benchmark** We introduce TSM-BENCH, a multilingual, multi-generator, and *multi-task* benchmark for task-specific MGT detection on Wikipedia, comprising 152,910 MGTs. We are among the first to study task-specific MGT detection. We plan to maintain this benchmark by adding more detectors, tasks, and languages. Code and data are available at GitHub.

- **Experiments** We evaluate 12 detectors from different model families, assess their generalisability across domains and tasks, and conduct a feature analysis to identify the linguistic cues detectors exploit.

- **Results** We demonstrate that (*i*) accuracy decreases by up to 32%, 20%, and 10% for zero-shot, off-the-shelf, and supervised detectors, respectively, compared to evaluations on generic data; and (*ii*) a **generalisation asymmetry** exists: models fine-tuned on task-specific data generalise to generic MGT both within and *across* domains, but not vice versa. Feature importance analysis shows that models trained on generic data overfit to LLM artefacts, exposing a limitation of prior benchmarks.

- **Implications** Compared with prior MGT benchmarks, our results suggest that earlier evaluations likely overestimated detector performance due to simplified generation settings. In practice, our findings indicate that most detectors cannot reliably support automated MGT detection in real-world contexts such as UGC platforms. TSM-BENCH therefore provides a valuable foundation for developing and evaluating more robust detectors, and we recommend training future models on a diverse combination of task-specific data.

## 2 RELATED WORK

**Wikipedia Editing Tasks** Wikipedia articles consist of a lead section, a tabular infobox, and a body organised into sections. Their content is written and maintained by volunteer editors who perform a wide range of tasks (Johnson et al., 2024). *Paragraph Writing* involves generating new encyclopaedic content, which is central to expanding knowledge on Wikipedia. Research has focused on expanding Wikipedia with agentic systems (Shao et al., 2024) or through RAG (Zhang et al., 2024b). *Summarisation* refers to producing the lead section of the article body, which introduces its most important points.[4] The literature treats lead section generation either as a multi-document (Liu et al., 2018; Gholipour Ghalandari et al., 2020; Hayashi et al., 2021) or a single-document (Gao et al., 2021; Perez-Beltrachini & Lapata, 2021) summarisation task. *Text Style Transfer* (TST) is the task of modifying the style of a sentence while preserving its meaning (Toshevska & Gievska, 2022). On Wikipedia, maintaining a Neutral Point of View[5] (NPOV) is a core content policy requiring all content to be written from a neutral perspective. Pryzant et al. (Pryzant et al., 2020) introduce the Wikipedia Neutrality Corpus (WNC), a large-scale collection of biased and neutralised sentence pairs retrieved from NPOV-related revisions.

**MGT Detection Benchmarks** Extensive work has benchmarked MGT detectors across domains, languages, and generators (Wu et al., 2025a). TuringBench (Uchendu et al., 2021) is one of the first benchmarks to study the Turing test and authorship attribution, using multiple generators in the news domain. MULTITuDE (Macko et al., 2023) expands MGT data beyond English, testing detectors in multilingual settings. MAGE (Li et al., 2024) covers multiple domains, generators, and detectors, benchmarking across eight increasingly challenging detection scenarios. M4 (Wang et al., 2024b) comprehensively includes various generators, languages, and domains, while M4GT (Wang et al., 2024a) expands M4 by incorporating additional languages and introducing human–machine mixed detection. Alongside the release of detection datasets (Guo et al., 2023; Su et al., 2023b; Yu et al., 2025), recent work has increasingly focused on adversarial attacks to evade detectors (He et al., 2024; Wu et al., 2024; Zheng et al., 2025).

---

[4] https://en.wikipedia.org/wiki/Wikipedia:Manual_of_Style
[5] https://en.wikipedia.org/wiki/Wikipedia:Neutral_point_of_view

Compared to prior MGT detection benchmarks, our work is among the first to consider MGT arising from task-specific scenarios. Most existing work relies on generic prompts such as "Write an article about neural networks." We illustrate the use of such generic prompts in related work (Wang et al., 2024b;a; Li et al., 2024; Macko et al., 2023) in Appendix 3. In contrast, our benchmark uses task-specific prompts as defined in Section 3.1.

**AI-Assisted Editing on Wikipedia** Wikipedia has a long history of AI-assisted tools. For example, ORES (Halfaker & Geiger, 2020) provides edit and page-quality predictions using multiple independent classifiers. Descartes (Sakota et al., 2023) generates short descriptions for Wikipedia articles using mBART. The emergence of LLMs has prompted researchers to investigate their impact on Wikipedia. Both Reeves et al. (2025) and Huang et al. (2025) find no or only marginal evidence of an effect on user engagement. Brooks et al. (2024) attempt to identify MGT on Wikipedia and estimate that roughly 5% of new articles may be AI-generated.

This paper extends our previous work (Quaremba et al., 2025), in which we introduced two datasets (WikiPS and mWNC) in English, Portuguese, and Vietnamese. TSM-BENCH extends the benchmark by adding more tasks, detectors, and LLMs, and provides more extensive experiments including evaluations of SOTA detectors, domain and cross-task generalisation, and feature-importance analyses. This allows us to draw critical conclusions about the reliability of detectors on UGC platforms such as Wikipedia.

## 3 TSM-BENCH

TSM-BENCH is a multilingual, multi-generator, and *multi-task* MGT detection benchmark designed to reflect real-world, task-specific LLM-generated text on Wikipedia. Our tasks are empirically grounded in research on Wikipedia editors' perceived use cases for LLM-assisted editing (Ford et al., 2023; Zhou et al., 2025), ensuring their relevance to practical applications. Our benchmark corpus comprises 152,910 parallel human- and machine-written texts across tasks, languages, and generators. Appendix Table 4 presents dataset statistics.

### 3.1 TASK DEFINITIONS

**MGT Detection** We define MGT detection as a binary classification task. Given a dataset $\mathcal{D} = \{(x_i, y_i)\}_{i=1}^N$, each instance consists of a text $x_i$ and a label $y_i \in \{0, 1\}$, where 0 denotes a human-written text and 1 a machine-generated text. A detector learns a function $f : \mathcal{X} \to \mathbb{R}$ that assigns a real-valued score to each text $x \in \mathcal{X}$. Using a threshold $\tau$, the predicted label is defined as $\hat{y} = 1$ if $f(x) \geq \tau$ and $\hat{y} = 0$ otherwise.

**Task-specific Generation** Let $f_\theta$ denote a language model with parameters $\theta$ that produces a textual output $o$. Let $I = \{i_1, \ldots, i_t\}$ be a set of detailed user instructions for $t$ natural language generation tasks, and let $C_t = \{c_1, \ldots, c_n\}$ be a set of contexts associated with task $t$. For example, $C_t$ may consist of retrieved evidence passages used to generate a new paragraph. We define **generic** generation as $o_{g_t} = f_\theta(g_t)$, where $g_t$ is an unconstrained, free-form prompt with *minimal* task instruction for task $t$. We define **task-specific** generation as $o_{\text{ts}} = f_\theta(i_t, C_t)$, where the model completes a constrained task using additional context $C_t$. This setting corresponds to the four Wikipedia editing tasks we investigate.

**Wikipedia Editing Tasks** We consider three editing tasks grounded in Ford et al. (2023) and Zhou et al. (2025), who survey Wikipedia editors about LLM-assisted editing practices. The three tasks are: ① **Paragraph Writing**, which involves generating new multi-sentence content or extending existing text. We define two subtasks: *Introductory Paragraph*, the task of writing the opening paragraph of a new section; and *Paragraph Continuation*, which extends an incomplete human-written paragraph. These subtasks allow us to test detection performance both on purely machine-written text and on text blending HWT and MGT. ② **Summarisation**, where the model generates a lead section of comparable length to a human-written reference, conditioned on the article's content. We frame this as a single-document abstractive summarisation task, following Wikipedia's Manual of Style[4] and prior work on Wikipedia summarisation (Gao et al., 2021; Perez-Beltrachini & Lapata, 2021). ③ **Text Style Transfer**, defined as *neutralising* revision-level NPOV violations (Pryzant

et al., 2020). We provide a biased sentence or paragraph as context and instruct the model to revise it in line with Wikipedia's neutrality guidelines. Focusing on NPOV violations ensures direct alignment with one of Wikipedia's core content policies.[6]

## 3.2 BENCHMARK CONSTRUCTION

**Data** We use WikiPS, a collection of paragraphs and summary–article pairs, and mWNC (Quaremba et al., 2025), an extension of WNC (Pryzant et al., 2020), as the human-written corpus, available in English, Portuguese, and Vietnamese. We randomly sample 2,700 HWT per task and language from the corresponding subsets. For the Paragraph Writing and Summarisation tasks, we balance each subset by length tertiles. For TST, we evaluate at the sentence level for all languages and at the paragraph level for English only, due to limited data in the other languages.

PROMPT EVALUATION

| Language | BLEU | RougeL | BERTScore | QAFactEval | Style Transfer |
|---|---|---|---|---|---|
| *Introductory Paragraph → RAG* | | | | | |
| English | 0.25 (+0.23) | 0.47 (+0.29) | 0.88 (+0.13) | 0.38 (+0.33) | - |
| Portuguese | 0.25 (+0.23) | 0.47 (+0.30) | 0.92 (+0.06) | 0.42 (+0.36) | - |
| Vietnamese | 0.30 (+0.26) | 0.55 (+0.23) | 0.92 (+0.07) | 0.36 (+0.30) | - |
| *Paragraph Continuation → RAG* | | | | | |
| English | 0.25 (+0.25) | 0.49 (+0.34) | 0.89 (+0.13) | 0.42 (+0.39) | - |
| Portuguese | 0.25 (+0.25) | 0.49 (+0.34) | 0.92 (+0.06) | 0.42 (+0.39) | - |
| Vietnamese | 0.32 (+0.30) | 0.57 (+0.26) | 0.92 (+0.07) | 0.38 (+0.34) | - |
| *Summarisation → One-shot* | | | | | |
| English | 0.17 (+0.11) | 0.36 (+0.10) | 0.83 (+0.04) | 0.46 (+0.01) | - |
| Portuguese | 0.11 (+0.05) | 0.29 (+0.06) | 0.88 (+0.01) | 0.47 (-0.01) | - |
| Vietnamese | 0.12 (+0.05) | 0.38 (+0.03) | 0.87 (+0.01) | 0.46 (+0.00) | - |
| *TST → Five-shot* | | | | | |
| English | 0.55 (+0.21) | 0.78 (+0.12) | 0.95 (+0.03) | - | 0.91 (+0.01) |
| Portuguese | 0.55 (+0.14) | 0.77 (+0.08) | 0.96 (+0.02) | - | 0.91 (+0.05) |
| Vietnamese | 0.55 (+0.12) | 0.78 (+0.05) | 0.96 (+0.01) | - | 0.84 (-0.01) |
| English P. | 0.55 (+0.21) | 0.78 (+0.12) | 0.95 (+0.03) | - | 0.96 (-0.01) |

Table 1: Prompt evaluation results. We report the highest-scoring prompt per language and task. For each metric, we present the highest score achieved, with the improvement over the baseline shown in percentage points (in parentheses).

For each task, we adapt prompts from the natural language generation literature shown to be most effective (Zhang et al., 2024a; Mukherjee & Dušek, 2024; Gao et al., 2024), and automatically evaluate them against a simple baseline. We conduct the evaluation on a length-stratified 10% sample of the target data using GPT-4o mini (Hurst et al., 2024), and select the highest-scoring prompt to generate MGT for our benchmark. Appendix B provides implementation details and prompt templates. The prompts we consider are as follows:

**Paragraph Writing** *Minimal* provides the generic baseline, instructing the model to write or continue a paragraph given article and section titles. *Content Prompts* extend Minimal by including up to ten content-related questions about the target HWT paragraph (e.g., "What are neural networks inspired by?"), generated using GPT-4o. *Naive retrieval-augmented generation* (RAG) (Gao et al., 2024) further augments Content Prompts with relevant retrieved content.

**Summarisation & TST** For Summarisation and TST, we use conceptually identical prompts. *Minimal* is a simple zero-shot prompt that instructs the model to summarise the article content/neutralise biased text. *Instruction* adds a detailed definition of the lead section/NPOV policy, alongside the baseline instructions to compile the lead section/neutralise the input text. *Few-shot* includes representative examples for each task in addition to the Instruction prompt.

---

[6] https://en.wikipedia.org/wiki/Wikipedia:Neutral_point_of_view

**Evaluation Metrics** We use BLEU (Papineni et al., 2002) and ROUGE (Lin, 2004) for n-gram overlap, and BERTScore (Zhang et al., 2020)[7] for semantic similarity. For Paragraph Writing and Summarisation, we assess factuality using QAFactEval (Fabbri et al., 2022).[8] For TST, we fine-tune pre-trained language models per language and report binary style classification accuracy.

**Evaluation** Table 1 presents our prompt evaluation results. For each task, we report the best-performing prompt, along with the percentage improvement over the baseline (in parentheses). Overall, prompts with richer context or more detailed instructions achieve greater gains across evaluation metrics: RAG substantially outperforms Minimal, while Few-shot prompts provide smaller improvements. The results demonstrate that task-specific MGT is of higher quality than generic MGT. For our benchmark, we select: **Paragraph Writing → RAG**, **Summarisation → One-shot**, and **TST → Five-shot**.

**MGT Generation** For each *task–language* subset, we generate MGT with six generators using their respective best-performing prompts. For LLMs, we use **GPT-4o** and **GPT-4o Mini** (Hurst et al., 2024), **Gemini 2.0 Flash** (Team et al., 2023), and **DeepSeek** (Guo et al., 2025). We also include two small language models (SLMs): **Qwen2.5-7B** (Yang et al., 2024a) and **Mistral-7B** (Jiang et al., 2023).

## 4 EXPERIMENTAL SETUP

We design five experiments to benchmark off-the-shelf, supervised, and zero-shot detectors, test their out-of-domain generalisability, analyse their behaviour through feature analysis, and evaluate cross-task transfer to inform future detector development. Further details on implementation, detectors, data, and training are provided in Appendix C.

**Experiment 1: Off-the-shelf detectors** We evaluate the performance of widely used off-the-shelf detectors on our tasks. For comparison, we also assess these detectors on generically generated Wikipedia articles, reflecting the setups of prior work (e.g., Guo et al., 2023; Macko et al., 2023; Li et al., 2024; He et al., 2024; Wang et al., 2024b;a; Wu et al., 2024). We generate these instances using the prompt *"Write a Wikipedia article about <title>"*. We select RADAR (Hu et al., 2023), Binoculars (Hans et al., 2024), Desklib, and e5-small (Dugan et al., 2024), as these models achieved the strongest performance on the RAID shared task (Dugan et al., 2024). We restrict this analysis to GPT-4o and English, as these detectors are primarily trained on English data.

**Experiment 2: Zero-shot and supervised detectors** We evaluate nine zero-shot and supervised detectors across each *task–language–generator* configuration. For each configuration, we fine-tune supervised models using hyperparameter search, and for zero-shot methods we calibrate the optimal classification threshold with Youden's $J$.

For supervised detectors, we use **XLM-RoBERTa** (Conneau et al., 2020) and **mDeBERTa** (He et al., 2023). As zero-shot white-box detectors, we include **Binoculars** (Hans et al., 2024), **LLR** (Su et al., 2023a), and **FastDetectGPT (White-Box)** (Bao et al., 2024). As zero-shot black-box detectors, we use **BiScope** (Guo et al., 2024), **Revise-Detect** (Zhu et al., 2023), **GECScore** (Wu et al., 2025b), and **FastDetectGPT (Black-Box)** (Bao et al., 2024).

**Experiment 3: Out-of-domain generalisation** We evaluate the generalisability of detectors trained on task-specific versus generic MGT, both *within* Wikipedia and *out-of-domain*. We consider two editor-driven domains where reliable MGT detection is equally important: social reviews (Yelp (Zhang et al., 2015) (EN), B2W (Real et al., 2019) (PT), ABSA (Nguyen et al., 2018) (VI)) and news (CNN/DM (Nallapati et al., 2016) (EN), Folha (Emdemor, 2023) (PT), News25 (Quang, 2022) (VI)). For all domains, we generate MGT using generic prompts.

---

[7]We use XLM-R (Conneau et al., 2020) as the backbone.

[8]As QAFactEval is English-only, we translated the Portuguese and Vietnamese prompt-evaluation sets using GPT-4o. Appendix D.1 shows that translation bias in this sample is minimal.

**Experiment 4: Feature analysis**    As our central argument concerns task-specific MGT, we investigate what models learn when trained on generic versus task-specific data. To this end, we train our best-performing model from Experiment 2 with the same configuration on each English dataset and compute Shapley Additive Explanations (SHAP) (Lundberg & Lee, 2017). SHAP values highlight which patterns models exploit to distinguish HWT from MGT.

**Experiment 5: Cross-task generalisation**    Different writing tasks may leave different traces of MGT. We examine how well detectors generalise across tasks by training a model on the full data of one task and evaluating it on all others. This experiment is crucial for understanding how to train future detectors for optimal performance.

We restrict Experiments 4 and 5 to GPT-4o and Qwen 2.5, using the best-performing model from Experiment 2. Given the parallel structure of our benchmark data, we report accuracy as our primary metric and additionally provide F1 scores for Experiment 2.

## 5    RESULTS

### 5.1    EXPERIMENT 1: OFF-THE-SHELF DETECTORS

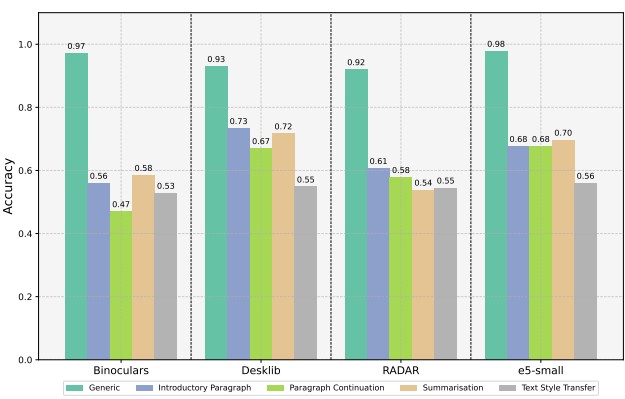

Figure 3: Comparison of off-the-shelf detectors on generic and task-specific MGT.

**Off-the-shelf detectors underperform on task-specific MGT.**    Figure 3 shows the accuracy of four off-the-shelf detectors on generic and task-specific data. All detectors achieve near-perfect accuracy of >93% on generic data, including the zero-shot method `Binoculars`. However, across tasks, accuracy drops to between 47% and 73%. This indicates that detectors which appear effective on generic data are likely to underperform in real-world scenarios where users rely on LLMs for specific tasks.

### 5.2    EXPERIMENT 2: ZERO-SHOT AND SUPERVISED DETECTORS

**Most detectors struggle to detect task-specific MGT.**    Table 2 reports detection results by task and language, averaged across the six generators.[9] Overall, all tasks pose challenges to detectors from every model family. Supervised models consistently outperform zero-shot methods, achieving average accuracies between 79.7% and 91.8% (excluding sentence-level TST), while zero-shot models fail to exceed 64.7% on average.

For Introductory Paragraph, supervised detectors achieve an average accuracy of 85.9% across languages, whereas white-box (57.7%) and black-box (62.3%) methods perform considerably worse. While both supervised detectors perform similarly, `Binoculars` achieves the highest average accuracy among white-box methods (61.8%), and `GECScore` leads among black-box methods (69.7%). For Paragraph Continuation, most zero-shot methods drop to near random-chance accuracy, with the exception of `BiScope`. This likely reflects blurred statistical disparities caused

---

[9]We report precision-recall curves for closer inspection of trained models in Appendix D.4.

| Detector | Introductory Paragraph | | | | | | Paragraph Continuation | | | | | |
|---|---|---|---|---|---|---|---|---|---|---|---|---|
| | English | | Portuguese | | Vietnamese | | English | | Portuguese | | Vietnamese | |
| | ACC | F1 | ACC | F1 | ACC | F1 | ACC | F1 | ACC | F1 | ACC | F1 |
| Binoculars | 57.8 | 60.4 | 61.3 | 61.8 | 66.4 | 67.3 | 52.1 | 41.7 | 55.8 | 55.0 | 60.6 | 60.0 |
| LLR | 50.9 | 63.2 | 52.6 | 56.0 | 55.4 | 30.9 | 50.6 | 23.5 | 51.6 | 45.2 | 52.2 | 25.1 |
| FDGPT (WB) | 53.6 | 43.8 | 58.4 | 60.6 | 63.0 | 56.5 | 50.7 | 36.7 | 54.5 | 49.1 | 57.8 | 40.3 |
| **Avg. White-box** | **54.1** | **55.8** | **57.4** | **59.4** | **61.6** | **51.6** | **51.1** | **34.0** | **54.0** | **49.8** | **56.8** | **41.8** |
| BiScope | 69.1 | 68.7 | 65.1 | 64.5 | 69.0 | 68.9 | 61.8 | 60.6 | 61.9 | 59.4 | 68.3 | 66.9 |
| Revise | 52.5 | 52.1 | 54.4 | 54.4 | 53.0 | 46.0 | 51.3 | 42.7 | 52.2 | 47.1 | 52.2 | 56.8 |
| GECScore | 75.6 | 74.0 | 70.8 | 72.3 | 62.7 | 62.8 | 56.1 | 54.7 | 55.0 | 40.8 | 52.8 | 36.9 |
| FDGPT (BB) | 53.6 | 41.7 | 58.7 | 55.3 | 62.2 | 56.8 | 51.3 | 17.2 | 55.7 | 41.6 | 59.1 | 43.1 |
| **Avg. Black-box** | **62.7** | **59.1** | **62.3** | **61.6** | **61.7** | **58.6** | **55.1** | **43.8** | **56.2** | **47.2** | **58.1** | **50.9** |
| xlm-RoBERTa | 84.6 | 84.2 | 82.2 | 81.3 | 85.5 | 84.7 | 84.3 | 84.2 | 85.2 | 84.8 | 88.1 | 88.0 |
| mDeBERTa | 89.0 | 88.8 | 83.8 | 83.1 | 86.3 | 85.6 | 86.2 | 86.1 | 86.3 | 86.2 | 87.4 | 87.4 |
| **Avg. Supervised** | **86.8** | **86.5** | **83.0** | **82.2** | **85.9** | **85.1** | **85.3** | **85.2** | **85.8** | **85.5** | **87.8** | **87.7** |

| Detector | Summarisation | | | | | | Text Style Transfer | | | | | | | |
|---|---|---|---|---|---|---|---|---|---|---|---|---|---|---|
| | English | | Portuguese | | Vietnamese | | English | | Portuguese | | Vietnamese | | English P. | |
| | ACC | F1 | ACC | F1 | ACC | F1 | ACC | F1 | ACC | F1 | ACC | F1 | ACC | F1 |
| Binoculars | 60.4 | 61.4 | 67.6 | 69.2 | 66.4 | 69.6 | 51.9 | 30.7 | 55.8 | 52.2 | 55.9 | 45.7 | 57.1 | 45.3 |
| LLR | 54.5 | 65.9 | 54.7 | 60.3 | 54.1 | 57.2 | 50.1 | 2.2 | 51.9 | 21.5 | 51.7 | 35.8 | 52.6 | 26.9 |
| FDGPT (WB) | 57.8 | 59.2 | 64.7 | 65.4 | 64.5 | 56.6 | 52.5 | 60.8 | 54.6 | 44.5 | 55.9 | 45.5 | 59.3 | 42.0 |
| **Avg. White-box** | **57.6** | **62.2** | **62.3** | **65.0** | **61.7** | **61.1** | **51.5** | **31.2** | **54.1** | **39.4** | **54.5** | **42.3** | **56.4** | **42.0** |
| BiScope | 70.7 | 69.9 | 68.0 | 66.0 | 70.5 | 70.2 | 57.3 | 56.9 | 60.3 | 59.6 | 59.6 | 58.3 | 57.3 | 56.9 |
| Revise | 54.0 | 56.3 | 53.3 | 57.0 | 53.0 | 57.9 | 55.1 | 60.0 | 53.3 | 54.1 | 56.1 | 60.1 | 57.4 | 58.0 |
| GECScore | 75.8 | 76.5 | 68.5 | 70.1 | 62.3 | 64.3 | 64.2 | 61.1 | 61.4 | 59.8 | 58.6 | 44.1 | 73.8 | 73.6 |
| FDGPT (BB) | 58.2 | 60.0 | 63.8 | 64.8 | 62.7 | 53.9 | 52.0 | 37.7 | 53.6 | 36.0 | 55.3 | 40.4 | 57.9 | 51.0 |
| **Avg. Black-box** | **64.7** | **65.6** | **63.4** | **64.5** | **62.1** | **61.6** | **57.1** | **53.9** | **57.1** | **52.4** | **57.4** | **50.7** | **61.6** | **59.9** |
| xlm-RoBERTa | 91.5 | 91.4 | 91.2 | 91.1 | 90.2 | 89.9 | 64.6 | 63.0 | 64.5 | 63.5 | 63.4 | 61.8 | 78.8 | 77.9 |
| mDeBERTa | 90.4 | 90.2 | 92.5 | 92.4 | 89.3 | 89.0 | 63.4 | 61.8 | 68.2 | 66.4 | 66.2 | 65.2 | 80.6 | 79.4 |
| **Avg. Supervised** | **90.9** | **90.8** | **91.8** | **91.8** | **89.8** | **89.5** | **64.0** | **62.4** | **66.4** | **64.9** | **64.8** | **63.5** | **79.7** | **78.7** |

Table 2: Detector accuracies (ACC) and F1-scores (F1) on task-specific MGT for each task and language, averaged across generators. English P. = English Paragraphs.

by the mixing of human and machine text, which undermines the signal on which these methods rely. We also observe a slight increase in average accuracy across detector families from English to Vietnamese, most pronounced for white-box detectors (e.g., `Binoculars`: 52.1 → 60). This linear trend occurs only in this task.

Summarisation yields the highest detection scores across model families and languages. Compared to Introductory Paragraph, both zero-shot families perform marginally better, while supervised models increase to or approach 90% accuracy across languages. Although most LLMs undergo extensive post-training on summarisation tasks, Wikipedia lead sections follow a distinctive style that provides strong cues for detectors. `BiScope` (69.7%) and `Binoculars` (64.8%) achieve the best performance within their respective families. Across languages, supervised detectors maintain consistent accuracy with a slight drop for Vietnamese; white-box methods perform best on Portuguese (62.3%), while black-box models reach the highest accuracy on English (64.7%).

For sentence-level TST, supervised detectors achieve an average accuracy of 65.1%, considerably lower than in the other tasks. Most zero-shot detectors perform only slightly above random chance across languages, with the notable exception of `GECScore`, which performs comparably to supervised detectors (e.g., 64.2% for English). We attribute the low detection scores in part to the sentence-level setting. When comparing English sentence- to paragraph-level data (English P.), we observe substantially higher average accuracies for the latter, most notably a 15.7% increase for supervised detectors.

## 5.3 EXPERIMENT 3: OUT-OF-DOMAIN GENERALISATION

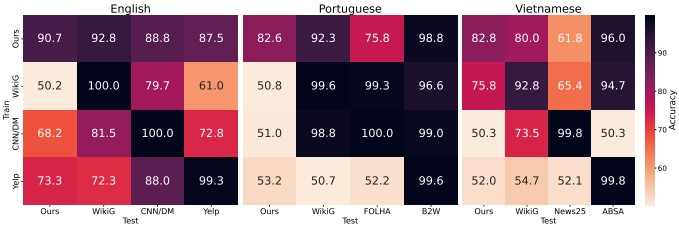

Figure 4: Out-of-domain accuracies of mDeBERTa by language with GPT-4o. Our dataset balances Introductory Paragraphs and Summarisation. WikiG (Wikipedia generic prompting), news (CN-N/DM, FOLHA, News25), and social reviews (Yelp, B2W, ABSA) represent generic MGT.

**Fine-tuning on task-specific data generalises to generic data within and across domains, but not vice versa.** Figure 4 shows confusion matrix accuracies by language for the best-performing model from our second experiment, mDeBERTa. We observe a **generalisation asymmetry**: when fine-tuned on task-specific data, mDeBERTa generalises well to generic data both within and *across* domains. However, when fine-tuned on generic data, the model fails to generalise to our data—even within the same domain. For example, mDeBERTa fine-tuned on our English data achieves an average accuracy of 89.7% across test sets (first matrix, first row). In contrast, when fine-tuned on generic data, no domain yields more than 73.3% test set accuracy on task-specific data. This pattern is consistent across most configurations, though most pronounced for English. Moreover, diagonal test set accuracies of 92.8–100% reinforce the findings of Experiments 1 and 2, underscoring that generic MGT is easy to detect *across domains*. Appendix Figure 10 reports results for Qwen 2.5, showing the same pattern.

## 5.4 EXPERIMENT 4: FEATURE ANALYSIS

**Fine-tuning on generic data tends to overfit to surface-level features.** To analyse the results of Experiments 1-3, we compare features learned by mDeBERTa when trained on generic versus task-specific English Wikipedia data. Figure 5 presents the five features with the highest SHAP values in each setting. mDeBERTa fine-tuned on our data assigns greater weight to semantically meaningful tokens, indicating stronger reliance on transferable MGT patterns. In contrast, when fine-tuned on generic data, SHAP values reveal heavy dependence on superficial cues such as section formatting (e.g., `"=="` or `"#"`).

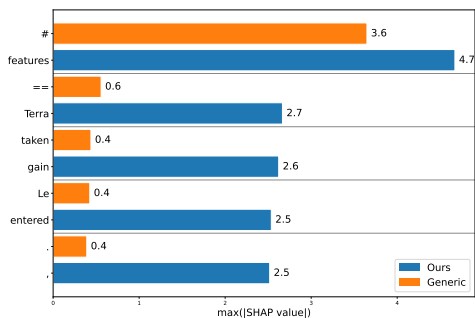

Figure 5: SHAP features for mDeBERTa.

## 5.5 EXPERIMENT 5: CROSS-TASK GENERALISATION

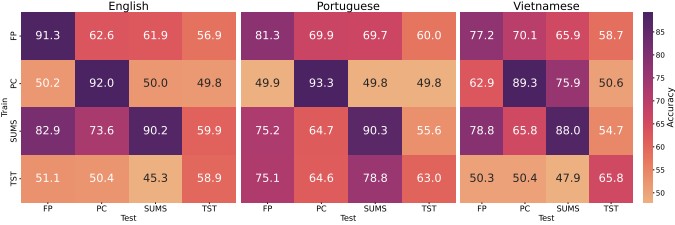

Figure 6: Cross-task accuracies of mDeBERTa by language with GPT-4o. IP = Introductory Paragraph, PC = Paragraph Continuation, SUMS = Summarisation, TST = Text Style Transfer.

**Cross-task performance is generally low.** Figure 6 presents cross-task accuracies by language for mDeBERTa. Overall, detection performance across tasks remains relatively low. For English, the average cross-task accuracy is 72.1% for Summarisation, compared with 60.5% for Introductory Paragraph and close to random chance for the other two tasks. The same trend holds for Portuguese and Vietnamese, as well as when using Qwen 2.5 (Appendix Figure 11). These results suggest that tasks exhibit distinct patterns that do not easily generalise across tasks. We therefore conclude that future detectors should be trained on a *combination* of different tasks.

## 6 DISCUSSION AND CONCLUSION

**Discussion** We find that most detectors struggle considerably on task-specific data. Through cross-domain experiments and feature analysis, we demonstrate that models trained on generic data tend to overfit to superficial MGT artefacts. This explains their strong in-domain but weak out-of-domain performance. In contrast to prior benchmarks (e.g. Guo et al., 2023; Macko et al., 2023; Li et al., 2024; He et al., 2024; Wang et al., 2024b;a; Wu et al., 2024), our results suggest that evaluations on generic data likely overestimate detector performance. This conclusion aligns with the findings of Doughman et al. (2025), who also highlight that classifier performance is often overestimated. Because we ground our task setup in observed editing practices of LLM usage (Ford et al., 2023; Zhou et al., 2025), we argue that most detectors are insufficient for supporting the automatic detection of MGT in real-world contexts. Recent work on adversarial attacks (He et al., 2024; Wu et al., 2024; Zheng et al., 2025) also reports reduced detection performance. However, our data are more challenging as they more realistically capture real-world *generation*, rather than relying on adversarial perturbations applied *post-generation*. For future work, we recommend developing and evaluating detectors on a diverse combination of common writing tasks. TSM-BENCH provides rich multilingual data to facilitate research in this direction. Future extensions could include additional languages, tasks, detectors, new domains, and analyses of the effects of combining tasks.

**Conclusion** We present TSM-BENCH, a multilingual, multi-generator, and *multi-task* benchmark for MGT detection, featuring diverse real-world LLM text generation tasks on Wikipedia. We show that most detectors underperform on task-specific MGT and highlight the limitations of evaluating detectors solely on generic MGT. Our findings suggest that existing benchmarks likely overestimate detector performance on UGC platforms and indicate that automatic MGT detection in real-world contexts remains unreliable.

**Limitations** First, we focus on three common editing tasks, although other equally important tasks exist (e.g., translation). We base these tasks on qualitative evidence of editors' LLM usage, although we cannot guarantee that all editors employ LLMs in these ways. Second, some of our style classifiers used for TST prompt evaluation perform poorly despite extensive fine-tuning. We address this limitation in Appendix C.2.1, but note that NPOV style classification remains challenging. Third, we stratify by length to avoid confounding effects, but we do not explore in detail how text length influences task-specific MGT detection. We leave this to future work.

## ETHICS STATEMENT

Our work uses publicly available content from Wikipedia, licensed under CC BY-SA. No private or sensitive information is included, and our experiments pose no risk to Wikipedia editors or the Wikipedias under study. Sensitive data about individual contributors are not identifiable or exposed in any way.

We obtain machine-generated data using four LLMs under their respective licences:

- GPT-4-mini: No specific license. OpenAI welcomes research publications.[10]
- Gemini 2.0: Apache 2.0[11]
- Qwen 2.0: Apache 2.0[12]

---

[10]https://openai.com/policies/sharing-publication-policy/
[11]https://github.com/google-gemini
[12]https://github.com/QwenLM/Qwen2.5

- Mistral: Apache 2.0[13]

All other datasets used in our work are publicly available. The license details are as follows:

- CNN/Daily Mail (News): Apache 2.0 License[14]
- FOLHA: CC0: Public Domain[15]
- News25: CC0: Public Domain[16]
- Yelp Reviews: Licensed under the Yelp Dataset License Agreement.[17] Permits usage for academic and non-commercial research purposes only.
- B2W-Reviews01: CC BY-NC-SA 4.0 License[18]
- VLSP 2018 ABSA: No specific license is provided, but the dataset is intended for research use.[19]

This study addresses limitations in previous evaluations of MGT detectors by assessing their performance within realistic editorial contexts. The objective is to provide more accurate and practical insights into the feasibility and utility of MGT detection in scenarios where humans employ LLMs in diverse ways to generate text for specific tasks. The experiments aim to inform the potential of MGT detectors as automated metrics or as tools to support users of UGC platforms in identifying machine-generated content.

**LLM Usage**   We use LLMs to correct spelling, grammar, and punctuation mistakes in our text. We do not copy and paste the corrected text but instead prompt for a detailed list of errors and incorporate them manually.

## REPRODUCIBILITY STATEMENT

We have taken several measures to ensure that our benchmark is reproducible and usable for future research. For data generation, we describe prompt design and evaluation in Section 3.2 and provide the full prompts alongside additional details on data generation in Appendix B. We outline our experimental setup in Section 4 and include further information in Appendix C, such as hyperparameter settings and hardware configurations. All experiments are run with a fixed random seed to guarantee full reproducibility of our results. We make all code and data publicly available.

## ACKNOWLEDGEMENTS

This work was supported by the Engineering and Physical Sciences Research Council [grant number EP/Y009800/1] through funding from Responsible AI UK (KP0011), and by UK Research and Innovation [grant number EP/S023356/1] through the UKRI Centre for Doctoral Training in Safe and Trusted Artificial Intelligence (www.safeandtrustedai.org).

---

[13]https://mistral.ai/news/announcing-mistral-7b
[14]https://huggingface.co/datasets/abisee/cnn_dailymail
[15]https://www.kaggle.com/datasets/marlesson/news-of-the-site-folhauol
[16]https://www.kaggle.com/datasets/haitranquangofficial/vietnamese-online-news-dataset
[17]Yelp Dataset Agreement
[18]https://github.com/americanas-tech/b2w-reviews01
[19]https://vlsp.org.vn/vlsp2018/eval/sa

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

# A  DATA

**MGT Generation Examples of Prior Work**  Table 3 presents the MGT generation prompts of three selected prior works, and an explanation of how they differ from TSM-BENCH.

| Paper | Prompt | Explanation |
|---|---|---|
| **M4GT-Bench** (Wang et al., 2024a) | Generate a Wikipedia summary for "`<title>`" (English). | Extends M4 by adding new *detection* but not generation tasks. The dataset expansion follows the generic generation paradigm, since summaries are produced without conditioning on article content, matching the generic setup described in Section 2.1. |
| **MAGE** (Li et al., 2024) | "Write a news article with the following headline: `<headline>`". | Uses a fully generic text-generation setup for the news domain. This matches the generic news prompt in our Experiment 3 used to construct generic news instances. |
| **MULTITuDE** (Macko et al., 2023) | "Write a news article in {language_name} [...]." | A multilingual MGT detection benchmark focused exclusively on generic news generation. It does not include task-specific settings and follows the same generic paradigm. |

Table 3: Examples of benchmarks relying on generic text-generation prompts, as defined in Section 3.1.

**Benchmark Statistics**  Table 4 presents benchmark statistics. We extend the data of our prior work (Quaremba et al., 2025) by adding two more generators and a new generation task, Paragraph Continuation. This increases the number of MGT samples by approximately half the size of the original data.

We performed data collection steps to mitigate the risk of MGT contamination. Specifically, all data was collected prior to the introduction of ChatGPT on 30 November 2022, we removed all text edited by bots, we considered only full articles rather than article stubs, and we included only text blocks containing at least one reference, ensuring a minimum standard of textual quality. While we cannot perfectly ensure that the data contains no MGT, we have taken these measures to minimise this risk.

| Corpus | Subset | Level | Language | Corpus N | Eval N | Experiment N | MGT N |
|---|---|---|---|---|---|---|---|
| **mWNC** | Text Style Transfer | Sentences | EN | 286,626 | 270 | 2,700 | 16,200 |
| | | | PT | 7,877 | 270 | 2,700 | 16,200 |
| | | | VI | 1,185 | 270 | 1,185 | 7,110 |
| | | Paragraphs | EN | 4,671 | 270 | 2,700 | 16,200 |
| **WikiPS** | Paragraph Writing | | EN | 96,860 | 270 | 2,700 | 16,200 |
| | | | PT | 72,965 | 270 | 2,700 | 16,200 |
| | | | VI | 98,315 | 270 | 2,700 | 16,200 |
| | Summarisation | | EN | 53,203 | 270 | 2700 | 16,200 |
| | | | PT | 36,075 | 270 | 2,700 | 16,200 |
| | | | VI | 45,500 | 270 | 2,700 | 16,200 |
| **Total** | | | | | **2,700** | **25,485** | **152,910** |

Table 4: TSM-Bench dataset statistics. Corpus N denotes the size of the data; Experiment N denotes the number of human-written texts; and MGT N denotes the total number of machine-generated texts.

**Language Selection Criteria** Table 5 presents details on our language selection. Our selection is guided by the goal of including high-, medium-, and low-resource languages, thereby informing communities beyond the English Wikipedia about the potential and limitations of AI detectors. Concretely, we selected languages that vary along two key dimensions. Firstly, the number of active editors. Secondly, we aimed to choose languages for which the resource ranking on Wikipedia aligns with the resource ranking in common LLM training corpora. To verify the latter, we relied on language distribution statistics from Common Crawl.

| Language | Resource Level | Active Users | CC Share (%) |
|---|---|---|---|
| English | High | 224,120 | 44.2668 |
| Portuguese | Medium | 8,343 | 2.1796 |
| Vietnamese | Low | 3,686 | 1.0326 |

Table 5: Overview of languages, resource levels, active Wikipedia user counts, and share language in Common Crawl data (CC-MAIN-2025-43).

## B  TASK DESIGN DETAILS

**Content Prompts** We model editors' LLM-assisted content generation through Content Prompts. This prompt variant is motivated from the literature on Wikipedia article generation (Shao et al., 2024), which models the process as information-seeking behaviour guided by asking questions. The underlying idea is that editors construct content by iteratively posing questions about the subject. Our Content Prompts are designed to simulate this cognitive process.

For instance, an editor aiming to expand a Wikipedia article might prompt a model to generate a paragraph in response to factual questions about a specific topic (e.g., "What is the difference between supervised and unsupervised learning?"" or "What is reinforcement learning used for?"), within a given section. For each human-written paragraph in our dataset, we prompt GPT-4 to generate a minimum of five content prompts for low-tertile paragraphs, and eight for medium- and high-tertile paragraphs. Although this method does not exhaustively cover all factual content from the HWT, it substantially improves the alignment of factual information between HWT and MGT.

When generating these prompts, a valid concern is that the resulting questions may contain hallucinations, which would deteriorate the quality of the generated texts. We rely on supportive evidence from fact-checking literature (Chen et al., 2022; Min et al., 2023). Min et al. (2023), which uses LLMs to generate atomic questions, finds that such questions are "effective and close to human," consistent with findings from prior work (Chen et al., 2022). Additionally, as reported in Table 11 in the Appendix, we show that text generated via Content Prompts leads to significant gains in our evaluation metrics, particularly in the factuality metric QAFactEval. Finally, our Content Prompts include detailed instructions on how to generate questions, which helps minimise the risk of irrelevance and hallucination.

**Naive RAG** We implement a web-based Naive RAG setup to reflect an editing scenario in which an editor, in addition to providing task instructions and content prompts, also supplies relevant context to minimise factual inaccuracies. Our RAG pipeline follows the indexing, retrieval, and generation modules of the Naive variant (Gao et al., 2024), with two key modifications: we prepend the pipeline with a Content Prompts and Web Search modules.

*Content Prompts and Web Search* For each paragraph, we generate diverse content prompts as described above. Each content prompt is used to query the Google Custom Search API,[20] retrieving the top 10 most relevant URLs. This results in a minimum of 80 URLs for low-tertile paragraphs and at least 100 URLs for medium- and high-tertile paragraphs.

*Indexing* We download the raw HTML of each scrappable web page and apply a series of preprocessing and cleaning steps. Each page is then split into chunks using LangChain's RecursiveChar-

---

[20]https://developers.google.com/custom-search/v1/overview

acterTextSplitter.[21] We compute BGE-M3[22] embeddings for each chunk and store them in a vector database.

*Retrieval and Generation* Each content prompt is treated as a query, for which we compute an embedding and retrieve the two most similar chunks from the vector database based on cosine similarity. These retrieved chunks are appended to the content prompt as context, guiding the model's generation. For the Paragraph Continuation task, we apply RAG only to the second half of each.

**In-context Learning** For Summarisation, we include 1–3 high-quality lead–content pairs retrieved from the respective Wikipedia Featured Articles page.[23] For TST, we include 1–5 randomly sampled biased–neutralised examples.

### B.1 PARAGRAPH WRITING PROMPTS

For brevity, we present prompts in English only.

INTRODUCTORY PARAGRAPH PROMPTS

**Minimal**

```
Please write the first paragraph for the section "{section_title}" in the
    Wikipedia article "{page_title}" using no more than {n_words} words.
    Only return the paragraph.
```

**Content Prompts**

```
Please write the first paragraph for the section "{section_title}" in the
    Wikipedia article "{page_title}".

Address the following key points in your response:
{content_prompts}

Use no more than {n_words} words. Only return the paragraph.
```

**RAG**

```
Use the following context to ensure factual accuracy when writing:
{context}

--

Please write the first paragraph for the section "{section_title}" in the
    Wikipedia article "{page_title}".

Address the following key points in your response:
{content_prompts}

Use the context above to inform your response, in addition to any
    relevant knowledge you have. Use no more than {n_words} words. Only
    return the paragraph in {language}.
```

PARAGRAPH CONTINUATION PROMPTS

**Minimal**

```
Please continue writing the following paragraph for the section "{
    section_title}" in the Wikipedia article "{page_title}".
```

---

[21]LangChain RecursiveCharacterTextSplitter documentation

[22]https://huggingface.co/BAAI/bge-m3

[23]https://en.wikipedia.org/wiki/Wikipedia:Featured_articles

```
Existing paragraph: "{p_first}"

Use no more than {n_words} words. Please only return the continuation of
    the paragraph.
```

**Content Prompts**

```
Please continue writing the following paragraph for the section "{
    section_title}" in the Wikipedia article "{page_title}".

Existing paragraph: "{p_first}"

Make sure that the continuation addresses these key points:
{content_prompts}

Use no more than {n_words} words. Please only return the continuation of
    the paragraph.
```

**RAG**

```
Use the following context to ensure factual accuracy when writing:
{context}

---

Please continue writing the below paragraph for the section "{
    section_title}" in the Wikipedia article "{page_title}".

Make sure that the continuation addresses these key points:
{content_prompts}

Existing paragraph:

"{trgt_first}"

Use the context above to inform your response, in addition to any
    relevant knowledge you have. Use no more than {trgt_n_toks} words.
    Only return the continuation of the paragraph in {language}.
```

## B.2  SUMMARISATION PROMPTS

**Minimal**

```
Your task is to summarize the below article with no more than {
    n_toks_trgt} words. Article:

"""{src}"""
```

**Instruction/Few-Shot**

```
Your task is to summarize an article to create a Wikipedia lead section.
- In Wikipedia, the lead section is an introduction to an article and a
    summary of its most important contents.
- Apart from basic facts, significant information should not appear in
    the lead if it is not covered in the remainder of the article.

Generate the lead for the article titled "{page_title}" using the article
    's body above with no more than {n_toks_trgt} words. Article:

"""{src}"""
```

### B.3 TST PROMPTS

**Minimal**

```
Please make this sentence/paragraph more neutral. **Make as few changes
    as possible and use no more than {trgt_n_words} words for the
    neutralised sentence/paragraph.** Sentence/Paragraph:

"""{src}"""
```

**Instruction/Few-Shot**

```
Please edit this biased Wikipedia sentence/paragraph to make it more
    neutral, aligning with Wikipedia's neutral point of view policy:

Achieving what the Wikipedia community understands as neutrality means
    carefully and critically analyzing a variety of reliable sources and
    then attempting to convey to the reader the information contained in
    them fairly, proportionately, and as far as possible without
    editorial bias. Wikipedia aims to describe disputes, but not engage
    in them. The aim is to inform, not influence. Editors, while
    naturally having their own points of view, should strive in good
    faith to provide complete information and not to promote one
    particular point of view over another. The neutral point of view does
     not mean the exclusion of certain points of view; rather, it means
     including all verifiable points of view which have sufficient due
     weight. Observe the following principles to help achieve the level of
      neutrality that is appropriate for an encyclopedia:

- Avoid stating opinions as facts.
- Avoid stating seriously contested assertions as facts.
- Avoid stating facts as opinions.
- Prefer nonjudgmental language.
- Do not editorialize.
- Indicate the relative prominence of opposing views.

**Make as few changes as possible and use no more than {trgt_n_words}
    words for the neutralised sentence/paragraph.** Output only the
    neutralized sentence/paragraph. Sentence/Paragraph:

"""{src}"""
```

## C  ADDITIONAL EXPERIMENTAL DETAILS

### C.1  DETECTOR DETAILS AND IMPLEMENTATIONS

We follow the taxonomy for detecting MGT proposed by (Yang et al., 2024b), which categorises detectors into three types: 1) zero-shot, 2) training-based, and 3) watermarking, although we exclude the latter from our experiments. The taxonomy further divides zero-shot methods into white-box and black-box, depending to whether the detector has access to the generator's logits other model internals. For all zero-shot models, when the originally used backbone LLM does not support one of our languages, we replace it with a multilingual model of the comparable size.

ZERO-SHOT WHITE-BOX

**LLR** (Su et al., 2023a) The Log-Likelihood Log-Rank Ratio (LLR), intuitively leverages the ratio of absolute confidence through log-likelihood to relative confidence through log rank about a sequence. We implement this detector with Bloom-3B.[24]

---

[24]https://huggingface.co/bigscience/bloom-3b

**Binoculars** (Hans et al., 2024) Binoculars introduces a metric based on the ratio of perplexity to cross-perplexity, where the latter measures how surprising the next-token predictions of one model are to another. We implement this detectors using Qwen2.5-7B[25] for the observer model and and Qwen2.5-7B-Instruct[26] for the performer model.

**FastDetectGPT White-Box** (Bao et al., 2024) DetectGPT (Mitchell et al., 2023) exploits that MGT tends to be located at negative curvature regions of the log probability function, from which a curvature-based detection criterion is defined. FastDetectGPT (WB) is an optimised version of DetectGPT that builds on the *conditional* probability curvature. We implement the white-box version with Bloom-3B.[24]

Zero-shot Black-box

**BiScope** (Guo et al., 2024) BiScope measures cross-entropy losses beywo output logits and original token and between output logitsaand the preceding input token. From statistics of these losses, they train a classifier to predict whether the text is machine-generated. We implement this detector as in the original paper with Llama 2-7B (Touvron et al., 2023).

**Revise** (Zhu et al., 2023) Revise builds on the hypothesis that ChatGPT[27] performs fewer revisions when generating MGT, and thus bases its detection criterion on the similarity between the original and revised articles. We implement this detector as in the original paper with GPT-3.5-turbo.[28]

**GECScore** (Wu et al., 2025b) Grammar Error Correction Score assumes that HWT contain more grammatical errors and calculates a Grammatical Error Correction score. We implement this detectors as in the original paper with GPT-3.5-turbo.[28]

**FastDetectGPT Black-Box** (Bao et al., 2024) In the black-box version, the scoring model is different from the reference model. We use BLOOM-3B for the reference model and BLOOM-1.7B for the scoring model.

Supervised

**XLM-RoBERTa** (Conneau et al., 2020) XLM-RoBERTa[29] is the multilingual version of RoBERTa (Liu et al., 2019) for 100 languages. RoBERTa is an improved version of BERT (Devlin et al., 2019) through more and longer training and dynamic masking modelling.

**mDeberTaV3** mDeberTaV3[30] is the multilingual version of DeBERTa (He et al., 2023) which improves BERT and RoBERTa through disentangled attention and enhanced mask decoder.

## C.2 Experimental Setups

### C.2.1 TST Style Classifiers

We fine-tune four style classifiers: one for each language at the sentence level, and an additional classifier for English at the paragraph level. The hyperparameter settings are provided in Table 6.

For English, we adopt the hyperparameters from the best-performing neutrality classifier available on Hugging Face.[31] As the English data contain nearly a quarter million English sentence pairs, we conduct fine-tuning on a smaller subset of the most recent 150k pairs, specifically filtered to include the keyword *NPOV* in the revision content, in order to further enhance precision. For Portuguese,

---

[25] https://huggingface.co/Qwen/Qwen2.5-7B
[26] https://huggingface.co/Qwen/Qwen2.5-7B-Instruct
[27] https://openai.com
[28] https://platform.openai.com/docs/models/gpt-3.5-turbo
[29] https://huggingface.co/FacebookAI/xlm-roberta-base
[30] https://huggingface.co/microsoft/mdeberta-v3-base
[31] https://huggingface.co/cffl/bert-base-styleclassification-subjective-neutral

| Language/Level | Models | Learning Rate | Batch Sizes | Epochs | Weight Decay |
|---|---|---|---|---|---|
| EN/Sent. | roberta-base | 1e-6 | 32 | 15 | 0.01 |
| PT/Sent. | xlm-roberta-base, mBERT | 5e-5, 1e-5, 5e-6 | 16, 32 | 2, 5, 8 | 0, 0.01 |
| VI/Sent. | xlm-roberta-base, mBERT | 5e-5, 1e-5, 5e-6, 1e-6 | 16, 32 | 2, 4, 6 | 0, 0.01 |
| EN/Para. | roberta-base | 5e-5, 1e-6, 5e-6 | 16, 32 | 3, 6, 9 | 0, 0.01 |

Table 6: Style Classifier Hyperparameter Settings.

we apply commonly used hyperparameter values, while for Vietnamese and English paragraphs, we extend the search space, as initial experiments yielded low detection performance.

| Level | Language | Pairs | Test Accuracy |
|---|---|---|---|
| Sentences | English | 300,000 | 73% |
| | Portuguese | 5738 | 63% |
| | Vietnamese | 2370 | 58% |
| Paragraphs | English | 9342 | 58% |

Table 7: Style Transfer Classifier Performance. Pairs denote biased and neutralised samples.

Table 7 reports the style classifier hyperparamter fine-tuning results. While fine-tuned models for English and Portuguese sentences yield satisfactory results, style accuracy for English paragraphs and Vietnamese sentences is low. In the following, we provide a qualitative analysis of both subsets and explain how we address these low performances.

**Low Style Classifier Performance Analysis**   Table 8 presents two representative examples of NPOV revisions from each subset. The first example in each case illustrates a clear NPOV violation. For instance, the phrase "considered the best footballer" in Vietnamese and "not as strong" in English are both subjective. However, as illustrated with the second examples, NPOV filtering also captures revisions related to political or historical content, which often rely on (subjectively) factual corrections rather than systematic semantic cues.

As we observed this pattern consistently across both subsets, we conducted additional data processing and hyperparameter tuning for the classifiers. We explored several strategies, including: (1) extending the list of NPOV-related keywords, (2) allowing multiple edit chunks per revision, (3) permitting multi-sentence edits within a single chunk, and (4) expanding the range of hyperparameter settings and model types. However, none of these approaches significantly improved style classifier performance.

Therefore, we selected the configuration that yielded the highest precision, adopting a conservative approach to extract NPOV-relevant revision pairs. Despite the relatively low classifier accuracy, we are confident that our dataset includes a high proportion of true positives.

### C.2.2   EXPERIMENT 2: ZERO-SHOT AND SUPERVISED DETECTORS

We fine-tune both training-based models per task and language on an 80/10/10 split with the hyperparameter choices displayed in Table 9.

| Hyperparameter | Values |
|---|---|
| Batch Size | 16, 32 |
| Learning Rate | 1e-5, 5e-6, 1e-6 |
| Epochs | 3, 5 |
| Seed | 42 |
| Resource | 1x NVIDIA A100 40GB |

Table 9: Hyperparameter settings for supervised-detectors.

| Subset | Biased Examples |
|---|---|
| Vietnamese | *c coi là cu th xut sc nht th gii và là cu th vĩ i nht mi thi i (**Greatest of All Time - GOAT**), Ronaldo là ch nhân ca 5 Qu bóng vàng châu Âu vào các năm 2008, 2013, 2014, 2016, 2017 và cũng là ch nhân 4 Chic giày vàng châu Âu, c hai u là k lc ca mt cu th châu Âu cùng nhiu danh hiu cao quý khác.* (EN: Considered the best football player in the world and the greatest of all time (GOAT), Ronaldo has won 5 Ballon d'Or awards in the years 2008, 2013, 2014, 2016, and 2017, as well as 4 European Golden Shoes—both records for a European player—along with many other prestigious titles.) |
| | *Ông tng phc v Lý Hoài Tiên, tng di quyn **nghch tc** S T Minh ca Ngy Yên.* (EN: He once served Lý Hoài Tiên, a general under the command of the rebel S T Minh of Ngy Yên.) |
| English Paragraphs | *He is **not as strong**, although still an exceptional warrior. Agamemnon clearly has a stubborn streak that **one can argue makes him even more arrogant** than Achilles. Although he takes few risks in battle, Agamemnon **still accomplishes great progress** for the Greeks.* |
| | *The population of Bangladesh ranks seventh in the world, but its area of approximately is ranked **ninety-fourth**, making it one of the most densely populated countries in the world, or the most densely populated country if small island nations and city-states are not included. It is the third-largest Muslim-majority nation, **but** has a smaller Muslim population than the Muslim minority in India. Geographically **dominated** by the fertile Ganges-Brahmaputra Delta, the country has annual monsoon floods, and cyclones are frequent.* |

Table 8: NPOV Revision Examples. Parentheses contain English translations. Highlighted words indicate words that were edited.

For zero-shot detectors, we use either a single NVIDIA A100 80GB or two NVIDIA A100 40GB GPUs. All models are trained on an HPC cluster.

### C.2.3 EXPERIMENT 3: GENERALISATION

For each language–generator–domain combination, we randomly sample 2,700 HWT from each dataset to generate generic MGT instances. These data reflects MGT genera setups of prior work (e.g., Guo et al., 2023; Macko et al., 2023; Li et al., 2024; He et al., 2024; Wang et al., 2024b;a; Wu et al., 2024) (see prompt templates below). For task-specific MGT, we randomly sample an equal number of instances from our Introductory Paragraph and Summarisation tasks, excluding text of the lowest length tertile.

Due to the open-ended nature of the test data, we truncate all outputs including our texts to 160 tokens to ensure comparable text lengths. All detectors are trained on the full training set ($N = 2,700$) and evaluated on 300 randomly drawn instances. We use the same hyperparameter settings as for Experiment 2 (see Table 9).

We run this experiment on a single NVIDIA A100 40GB.

#### TEST DATA

We consider two additional domains—news and social reviews—for which reliable MGT detection is equally important as on Wikipedia.

#### WIKIPEDIA

For all three languages, we randomly sample from our base WikiPS dataset to create full articles consisting of the lead section and article body with minimal formatting.

NEWS

**CNN/DM**   CNN/Daily Mail (Nallapati et al., 2016) is an English dataset containing over 300,000 news articles from CNN and the Daily Mail, each paired with a summary composed of bullet-pointed highlight sentences. We use only the full article text in our experiments.

**FOLHA**   Folha de São Paulo (FOLHA) (Emdemor, 2023) is a large-scale collection of 167,053 news titles and articles from the Brazilian newspaper of the same name. The dataset covers the period from January 2015 to September 2017.

**News25**   The Vietnamese Online News Dataset (News25) (Quang, 2022) is a large-scale collection of over 150,000 news articles from the 25 most popular Vietnamese news sites, collected in July 2022. Each entry includes a title and the main article body, along with additional metadata.

SOCIAL REVIEWS

**Yelp**   The Yelp dataset (Zhang et al., 2015) is a large-scale collection of approximately 700,000 business reviews written on the Yelp platform. It covers businesses across eight metropolitan areas in the United States and Canada.

**B2W**   B2W-Reviews01 (Real et al., 2019) is a Portuguese dataset containing over 130,000 e-commerce customer reviews. The reviews were collected from the Americanas.com website between January and May 2018.

**ABSA**   The VLSP 2018 Aspect-Based Sentiment Analysis (ABSA) dataset (Nguyen et al., 2018) includes 4,751 restaurant reviews and 5,600 hotel reviews in Vietnamese. We consider only the restaurant domain, which consists of reviews collected from www.foody.vn.

PROMPT TEMPLATES

For brevity, we present prompts in English only.

WIKIPEDIA TST

```
Write a Wikipedia article with the title "{title}", the article should at
    least have 250 words.
```

**CNN/DM**

```
Write a news article given the following highlights: """{highlights}"""
```

**Yelp**

```
Given the first few words of the review, continue the review with a
    minimum of 20 words. Review beginning: "{beginning}"
```

## D   ADDITIONAL RESULTS

### D.1   TRANSLATION QUALITY

Table 10 reports a quality assessment of our back-translations using the DeepL API, one of the leading commercial translation systems. Across tasks and languages, standard machine translation metrics (BLEU, ROUGE, and BERTScore) indicate consistently high translation quality. This suggests that, while we cannot fully rule out translation bias, its impact on the factuality evaluation of Portuguese and Vietnamese prompts is likely minimal. This interpretation is further supported by the near-identical factuality scores across languages in Table 3.2. Finally, we emphasize that translation is used only for evaluating prompt factuality with QAFactEval, all detection experiments operate solely on the original source texts.

| Task (Language) | BLEU | ROUGE-1 | ROUGE-2 | BERTScore |
|---|---|---|---|---|
| Paragraphs (PT) | 71.50 | 0.7838 | 0.6468 | 0.9671 |
| Paragraphs (VI) | 67.95 | 0.8475 | 0.7164 | 0.9486 |
| Summarization (PT) | 74.24 | 0.8767 | 0.8013 | 0.9736 |
| Summarization (VI) | 63.46 | 0.8868 | 0.7520 | 0.9411 |

Table 10: Back-translation quality metrics (BLEU, ROUGE, and BERTScore) across tasks and languages.

## D.2 LINGUISTIC DESCRIPTIVE ANALYSIS

Figures 7 and 8 show the results of the linguistic descriptive analysis for Portuguese and Vietnamese, respectively.

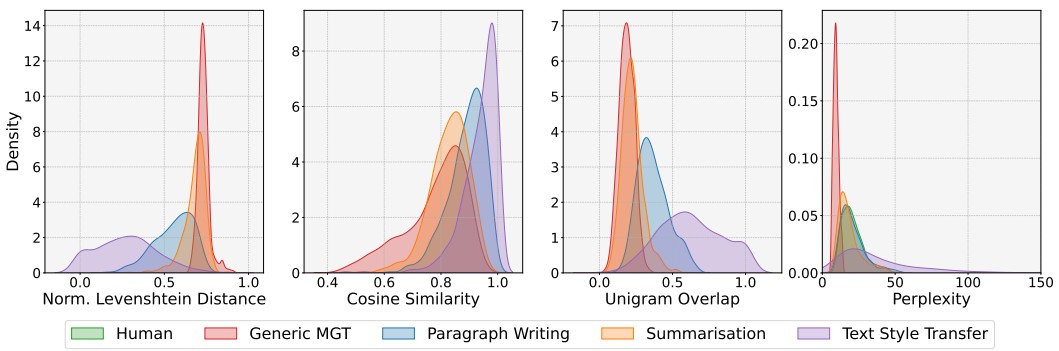

Figure 7: Comparison of textual characteristics between human, generic, and our task-specific MGT in Portuguese.

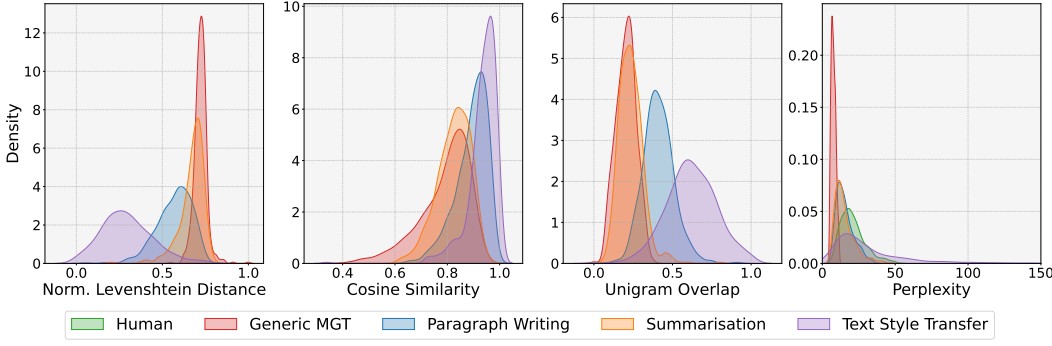

Figure 8: Comparison of textual characteristics between human, generic, and our task-specific MGT in Vietnamese.

## D.3 PROMPT EVALUATION

| Language | Technique | BLEU | ROUGE-1 | ROUGE-2 | ROUGE-L | BERTScore | QAFactEval |
|---|---|---|---|---|---|---|---|
| *Introductory Paragraph* | | | | | | | |
| English | Minimal | 0.02 | 0.29 | 0.06 | 0.17 | 0.76 | 0.06 |
| | Content Prompts | 0.22 | 0.57 | 0.31 | 0.44 | **0.88** | 0.25 |
| | RAG | **0.25** | **0.61** | **0.35** | **0.47** | **0.88** | **0.38** |
| Portuguese | Minimal | 0.02 | 0.31 | 0.06 | 0.17 | 0.86 | 0.06 |
| | Content Prompts | 0.20 | 0.56 | 0.30 | 0.41 | 0.91 | 0.25 |
| | RAG | **0.25** | **0.61** | **0.37** | **0.47** | **0.92** | **0.42** |
| Vietnamese | Minimal | 0.04 | 0.67 | 0.26 | 0.32 | 0.85 | 0.06 |
| | Content Prompts | 0.28 | 0.78 | 0.52 | 0.54 | 0.91 | 0.27 |
| | RAG | **0.30** | **0.79** | **0.54** | **0.55** | **0.92** | **0.36** |
| *Paragraph Continuation* | | | | | | | |
| English | Minimal | 0.01 | 0.24 | 0.03 | 0.15 | 0.75 | 0.03 |
| | Content Prompts | 0.21 | 0.58 | 0.32 | 0.45 | **0.88** | 0.30 |
| | RAG | **0.25** | **0.60** | **0.36** | **0.49** | **0.89** | **0.42** |
| Portuguese | Minimal | 0.01 | 0.25 | 0.04 | 0.15 | 0.86 | 0.03 |
| | Content Prompts | 0.20 | 0.57 | 0.32 | 0.44 | **0.92** | 0.27 |
| | RAG | **0.25** | **0.60** | **0.38** | **0.49** | **0.92** | **0.42** |
| Vietnamese | Minimal | 0.01 | 0.62 | 0.21 | 0.31 | 0.85 | 0.04 |
| | Content Prompts | 0.31 | **0.78** | **0.54** | 0.56 | **0.92** | 0.31 |
| | RAG | **0.32** | **0.78** | **0.54** | **0.57** | 0.02 | **0.38** |

Table 11: Paragraph Writing Prompts Evaluation Results.

Table 11 presents our prompting evaluation results. We find that our Naive RAG approach consistently outperforms both Minimal and Content Prompts across subtasks and languages. The low evaluation scores for Minimal prompts highlight that MGT produced in prior work is often synthetically divergent from its human-written references. While Content Prompts substantially improve performance, Naive RAG further enhances generation quality—particularly in terms of factual consistency, which is critical for encyclopedic content.[32] Based on these findings, we adopt Naive RAG as the prompting strategy for the paragraph writing task in our MGT detection experiments.

| Language | Technique | BLEU | ROUGE-1 | ROUGE-2 | ROUGE-L | BERTScore | QAFactEval |
|---|---|---|---|---|---|---|---|
| English | Minimal | 0.06 | 0.37 | 0.13 | 0.26 | 0.79 | 0.45 |
| | Instruction | 0.13 | 0.44 | 0.21 | 0.33 | 0.82 | **0.46** |
| | One-shot | **0.18** | **0.47** | **0.24** | 0.36 | 0.83 | **0.46** |
| | Two-shot | **0.18** | **0.47** | **0.24** | 0.36 | 0.83 | **0.46** |
| | Three-shot | 0.16 | 0.46 | 0.23 | 0.35 | **0.83** | **0.46** |
| Portuguese | Minimal | 0.06 | 0.35 | 0.13 | 0.23 | 0.87 | **0.48** |
| | Instruction | **0.11** | 0.42 | 0.19 | 0.30 | **0.88** | **0.48** |
| | One-shot | **0.11** | 0.42 | 0.19 | 0.29 | **0.88** | **0.48** |
| | Two-shot | **0.11** | **0.43** | 0.19 | **0.30** | **0.88** | 0.47 |
| | Three-shot | 0.12 | **0.43** | **0.20** | **0.30** | **0.88** | 0.47 |
| Vietnamese | Minimal | 0.07 | 0.63 | 0.28 | 0.35 | 0.86 | **0.45** |
| | Instruction | 0.11 | 0.64 | 0.31 | **0.38** | **0.87** | 0.43 |
| | One-shot | **0.12** | 0.65 | **0.32** | **0.38** | **0.87** | **0.45** |
| | Two-shot | **0.12** | **0.66** | **0.32** | **0.38** | **0.87** | 0.44 |
| | Three-shot | 0.11 | 0.65 | **0.32** | **0.38** | **0.87** | 0.42 |

Table 12: Summarisation Prompts Evaluation Results.

Table 12 presents the summarisation prompt evaluation results, showing that across languages, Instruction and Few-shot achieve higher overlap and semantic similarity scores, although Few-shot only marginally improves over Instruction. Factuality scores remain relatively stable across prompts, presumably because summarisation is a core task in aligning LLMs through reinforcement learning from human feedback (Ouyang et al., 2022). Given that increasing the number of shots does not yield further improvements, and considering the context window of smaller LLMs, we select one-shot prompting for our experiments.

---

[32]https://en.wikipedia.org/wiki/Wikipedia:Verifiability

| Language | Technique | BLEU | ROUGE-1 | ROUGE-2 | ROUGE-L | BERTScore | ST |
|---|---|---|---|---|---|---|---|
| English | Minimal | 0.35 | 0.68 | 0.52 | 0.66 | 0.92 | 0.90 |
| | Instruction | 0.36 | 0.68 | 0.52 | 0.66 | 0.92 | **0.94** |
| | One-shot | 0.52 | 0.78 | 0.65 | 0.76 | **0.95** | 0.91 |
| | Two-shot | 0.47 | 0.75 | 0.61 | 0.73 | 0.94 | 0.90 |
| | Three-shot | 0.54 | 0.79 | 0.67 | 0.78 | **0.95** | 0.89 |
| | Four-shot | **0.56** | **0.80** | **0.69** | **0.79** | **0.95** | 0.89 |
| | Five-shot | **0.55** | **0.80** | 0.68 | 0.78 | **0.95** | 0.91 |
| Portuguese | Minimal | 0.41 | 0.71 | 0.58 | 0.69 | 0.94 | 0.86 |
| | Instruction | 0.40 | 0.70 | 0.57 | 0.67 | 0.94 | 0.88 |
| | One-shot | 0.50 | 0.75 | 0.64 | 0.74 | **0.96** | 0.90 |
| | Two-shot | 0.51 | 0.77 | 0.65 | 0.75 | **0.96** | 0.89 |
| | Three-shot | 0.53 | 0.78 | 0.66 | 0.76 | **0.96** | 0.91 |
| | Four-shot | **0.58** | **0.81** | **0.70** | **0.79** | **0.96** | **0.92** |
| | Five-shot | 0.55 | 0.79 | 0.68 | 0.77 | **0.96** | 0.91 |
| Vietnamese | Minimal | 0.43 | 0.78 | 0.65 | 0.73 | 0.95 | 0.84 |
| | Instruction | 0.45 | 0.80 | 0.67 | 0.73 | 0.94 | 0.79 |
| | One-shot | 0.44 | 0.78 | 0.66 | 0.71 | 0.95 | **0.88** |
| | Two-shot | 0.51 | 0.82 | 0.70 | 0.76 | 0.95 | 0.87 |
| | Three-shot | 0.50 | 0.81 | 0.70 | 0.75 | 0.95 | 0.85 |
| | Four-shot | 0.51 | 0.82 | 0.70 | 0.76 | 0.95 | 0.85 |
| | Five-shot | **0.55** | **0.83** | **0.73** | **0.78** | **0.96** | 0.84 |
| English Para. | Minimal | 0.35 | 0.68 | 0.52 | 0.66 | 0.92 | 0.97 |
| | Instruction | 0.36 | 0.68 | 0.52 | 0.66 | 0.92 | **0.99** |
| | One-shot | 0.52 | 0.78 | 0.65 | 0.76 | **0.95** | 0.95 |
| | Two-shot | 0.47 | 0.75 | 0.61 | 0.73 | 0.94 | 0.98 |
| | Three-shot | 0.54 | 0.79 | 0.67 | 0.78 | **0.95** | 0.96 |
| | Four-shot | 0.56 | **0.80** | **0.69** | **0.79** | **0.95** | 0.95 |
| | Five-shot | 0.55 | **0.80** | 0.68 | 0.78 | **0.95** | 0.96 |

Table 13: TST Prompts Evaluation Results.

Table 13 presents the prompt evaluation metrics for the TST task, evaluated at the sentence level for all languages, and additionally at the paragraph level for English. Across languages and levels, we find that four- and five-shot prompting consistently outperforms Minimal and Instruction prompts. While differences in semantic similarity and style transfer are marginal across prompts, we observe substantial improvements in overlap-based metrics as the number of few-shot examples increases. These improvements can be attributed to the fact that neutralisation edits in mWNC tend to be relatively minimal. For instance, in the English sentence subset, on average only 14% of words are deleted and 7% added—similar trends hold for the other subsets. As a result, the model appears to learn from the examples to apply similarly sparse edits, thereby producing outputs that match the reference text more closely in terms of n-gram overlap. Based on these findings, we adopt five-shot prompting to generate MGT in our subsequent experiments.

## D.4   EXPERIMENT 2: PRECISION-RECALL CURVES ACROSS TASKS AND LANGUAGES

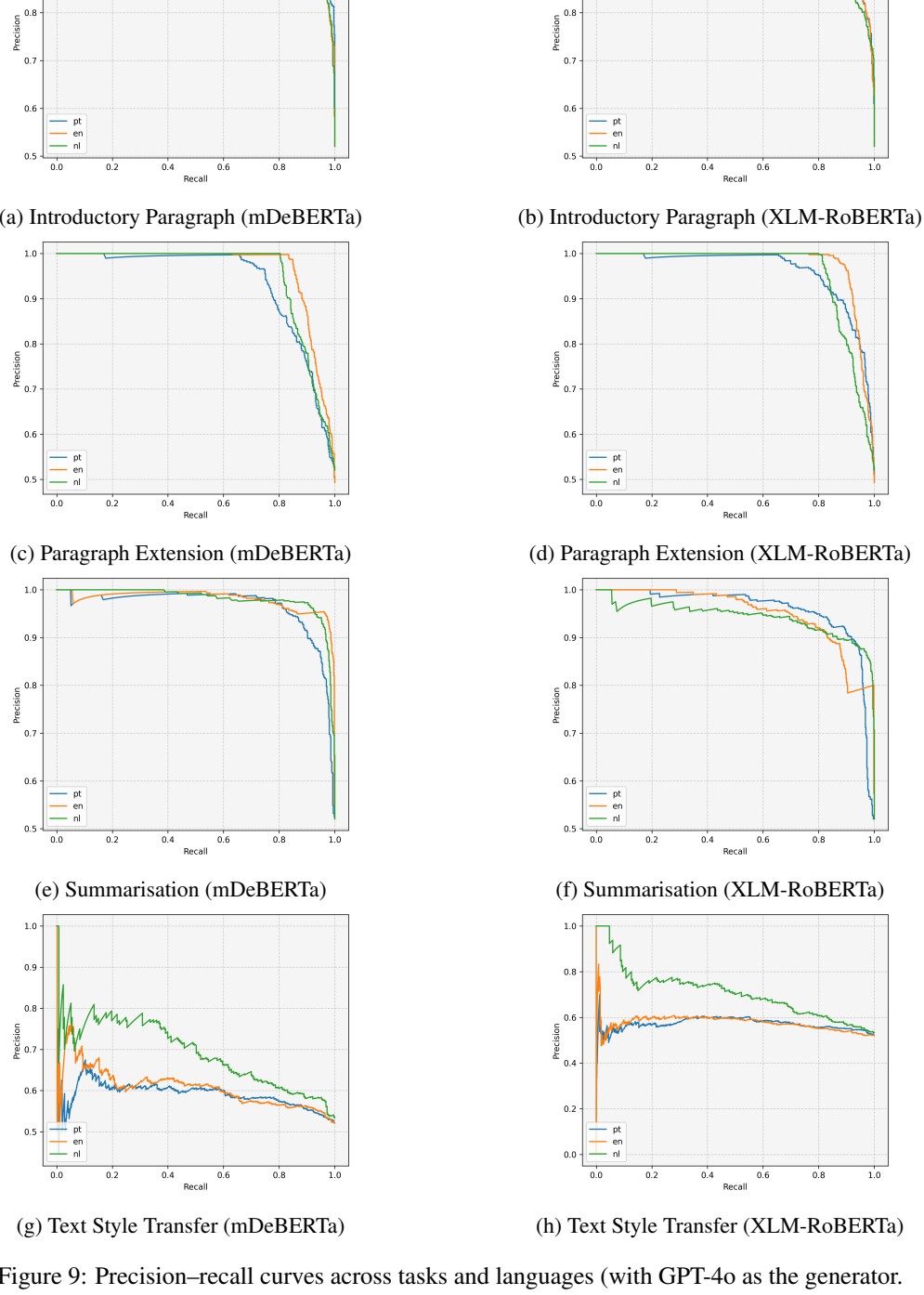

Figure 9: Precision–recall curves across tasks and languages (with GPT-4o as the generator.

## D.5 EXPERIMENT 2: RESULTS BY LANGUAGE, MODEL AND TASKS

Tables 14 and 15 present the full results of experiment 2.

Table 14: Detector accuracies (ACC) and F1-scores (F1) on task-specific MGT for each task, language, and generator.

| Detector | English Text Style Transfer | | | | | | | | | | | | | |
|---|---|---|---|---|---|---|---|---|---|---|---|---|---|
| | GPT-4o | | GPT-4o mini | | Gemini 2.0 | | DeepSeek | | Qwen 2.5 | | Mistral | | Avg | |
| | ACC | F1 | ACC | F1 | ACC | F1 | ACC | F1 | ACC | F1 | ACC | F1 | ACC | F1 |
| Binoculars | 0.70 | 0.62 | 0.58 | 0.53 | 0.55 | 0.47 | 0.50 | 0.19 | 0.52 | 0.39 | 0.57 | 0.51 | **0.57** | **0.45** |
| LLR | 0.60 | 0.71 | 0.52 | 0.25 | 0.51 | 0.22 | 0.50 | 0.00 | 0.50 | 0.03 | 0.53 | 0.40 | **0.53** | **0.27** |
| FDGPT (WB) | 0.80 | 0.78 | 0.60 | 0.63 | 0.56 | 0.60 | 0.50 | 0.00 | 0.52 | 0.60 | 0.58 | 0.61 | **0.59** | **0.54** |
| Avg (White-box) | 0.70 | 0.71 | 0.57 | 0.47 | 0.54 | 0.43 | 0.50 | 0.06 | 0.52 | 0.34 | 0.56 | 0.51 | **0.56** | **0.42** |
| BiScope | 0.55 | 0.56 | 0.55 | 0.55 | 0.58 | 0.57 | 0.66 | 0.65 | 0.56 | 0.55 | 0.54 | 0.53 | **0.57** | **0.57** |
| Revise | 0.80 | 0.80 | 0.53 | 0.62 | 0.52 | 0.56 | 0.54 | 0.54 | 0.53 | 0.42 | 0.52 | 0.55 | **0.57** | **0.58** |
| GECScore | 0.85 | 0.86 | 0.83 | 0.82 | 0.64 | 0.67 | 0.70 | 0.69 | 0.73 | 0.69 | 0.67 | 0.69 | **0.74** | **0.74** |
| FDGPT (BB) | 0.75 | 0.71 | 0.59 | 0.62 | 0.54 | 0.55 | 0.50 | 0.00 | 0.61 | 0.63 | 0.58 | 0.54 | **0.58** | **0.51** |
| Avg (Black-box) | 0.74 | 0.73 | 0.63 | 0.65 | 0.57 | 0.59 | 0.60 | 0.47 | 0.58 | 0.57 | 0.58 | 0.58 | **0.62** | **0.60** |
| xlm-RoBERTa | 0.76 | 0.74 | 0.78 | 0.77 | 0.78 | 0.78 | 0.91 | 0.91 | 0.78 | 0.77 | 0.71 | 0.71 | **0.79** | **0.78** |
| mDeBERTa | 0.82 | 0.81 | 0.83 | 0.83 | 0.77 | 0.76 | 0.92 | 0.92 | 0.81 | 0.81 | 0.67 | 0.64 | **0.81** | **0.79** |
| Avg (Supervised) | 0.79 | 0.77 | 0.81 | 0.80 | 0.78 | 0.77 | 0.92 | 0.92 | 0.80 | 0.79 | 0.69 | 0.67 | **0.80** | **0.79** |

Table 15: Detector accuracies (ACC) and F1-scores (F1) on task-specific MGT for TST of English paragraphs.

**Mistral Error Analysis** We observe anomalous evaluation metrics for Vietnamese texts written by Mistral. While both zero-shot detectors achieve random chance accuracy and often zero F1-scores, training-based detectors achieve almost perfect metrics. After checking the data, we observe that Mistral—in contrast to the other models—does not follow the instructions in our prompts. Typical errors include outputting text mid-sentence or returning English text, despite the last sentences of our prompt emphasizing to return text in Vietnamese. These flaws explain the strong performance of training-based detectors, as they pick up these syntactic imperfections, whereas zero-shot detectors seem unable to find clearly distinctive patterns based on model internals or token patterns.

## D.6 Experiment 3: Results for Qwen 2.5

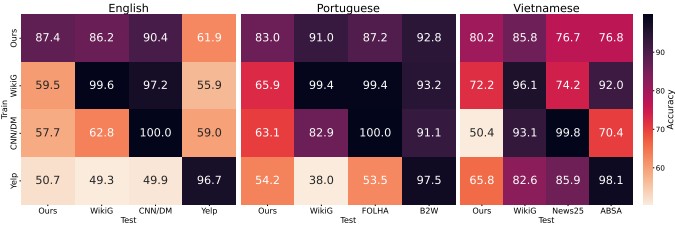

Figure 10: Out-of-domain accuracies of mDeBERTa by language with Qwen 2.5. Our dataset balances Introductory Paragraphs and Summarisations. WikiG (Wikipedia generic prompting), news (CNN/DM, FOLHA, News25), and social reviews (Yelp, B2W, ABSA), are generic MGT.

## D.7 Experiment 5: Results for Qwen 2.5

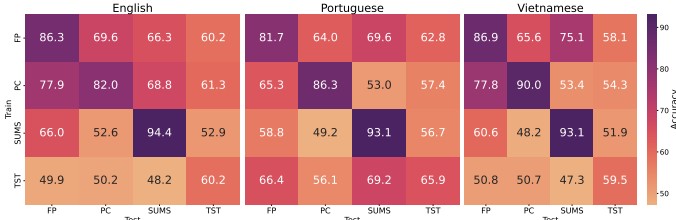

Figure 11: Cross-task domain accuracies of mDeBERTa by language with Qwen 2.5. IP=Introductory Paragraph, PC=Paragraph Continuation, SUMS=Summarisation, TST=Text Style Transfer.

