# OpenReview forum: "TSM-Bench: Detecting LLM-Generated Text in Real-World Wikipedia Editing Practices"
_ICLR.cc/2026/Conference — ICLR 2026 Poster_

### Official Review · Reviewer_mKYw · 2025-10-29

**Soundness:** 3
**Presentation:** 3
**Contribution:** 3
**Rating:** 8
**Confidence:** 3

**Summary:**

This paper proposes a multilingual, multi-generator, and multi-task benchmark for machine-generated text (MGT) detection on Wikipedia, encompassing 4 tasks, 152,910 examples and six text generators in total. An extensive set of experiments is conducted, including evaluation of 12 detectors from different methods (zero-shot, off-the-shelf, supervised) and model families, feature importance analysis, out-of-domain and cross-task generalisation. Results show degradation of metrics on task-specific generation data compared to evaluations on generic data, and generalisation asymmetry in favor of task-specific generation data. Importantly, the limitations of existing benchmarks and MGT detectors in real-world conditions are emphasized.

**Strengths:**

The paper identifies a critical gap in existing benchmarks, where inflated performance metrics often fail to translate to real-world applications. The authors propose a novel benchmark that is multilingual, incorporates multiple generators, and covers main tasks relevant to user-generated content platforms like Wikipedia. An underlying contribution is the original formulation dividing data generation into "task-specific" (constrained by context) and "generic" (free-form) categories. The experiments demonstrate a substantial performance decrease for all detector types on task-specific data and reveal a clear generalization asymmetry: task-specific training aids in detecting generic text but not the reverse. The feature importance analysis indicates that models trained on generic data tend to overfit to superficial cues.

The paper is well-structured, competently written, and backed by thorough analysis, including a prompt impact evaluation and an examination of performance gaps between different detector methods.

The public release of the code and data represents a meaningful contribution to the community, enabling reproducibility and future research, in particular the development and evaluation of MGT detectors.

**Weaknesses:**

1. The out-of-domain generalization experiment lacks transparency, as it conflates the effects of the domain and the generation conditioning (task-specific vs. generic). A more complete picture would require a fine-tuning experiment on the authors' own "Our" data using generic MGT to directly compare the performance impact of the training data's contextual conditioning generation.
2. The benchmark, though covering crucial real-world tasks, omits others like translation and creative writing, and the scope of domains remains limited to Wikipedia.
3. The paper misses an opportunity to provide examples of generic MGT cases and does not explore balancing task-specific and generic data for optimal detector training.
4. The quality of the LLM-based translation is not checked.

**Questions:**

1. Some details are unclear, such as the specific model used for BERTScore, the references for per-language fine-tuned models, and the definition of "Wiki IO." in Exp. 3.
2. The prompt evaluation relies on a single model, limiting the reliability of its conclusions.
3. The reliability of the RAG pipeline, particularly the relevance of retrieved chunks, is not verified.
4. The justification for the specific language selection is currently poor and needs strengthening.
5. Below are some suggestions to fix typos and text errors:
— L384 “accuracy of 89.7%”, in Appendix D.1 captions for figures are the same.
— Appendix A: “DATA SET STATISTICS” –> “BENCHMARK STATISTICS”; “Corpus denotes the size of the data” –> “Corpus denotes the source of the data”.
— Rephrase L070  “content integrity of UGC”.
— Formulation “generic generation” is not clear.
— The subsection "PROMPT EVALUATION" is missing a section number

---

> ### Author Response · Authors · 2025-11-20
>
> Thank you for your review and positive assessment of our work! Please find our responses to your comments below.
>
> _We will soon upload an updated manuscript that incorporates the suggestions addressed in our rebuttal._
>
> ---
> > Weakness 1: The out-of-domain generalization experiment lacks transparency, as it conflates the effects of the domain and the generation conditioning (task-specific vs. generic). A more complete picture would require a fine-tuning experiment on the authors' own "Our" data using generic MGT to directly compare the performance impact of the training data's contextual conditioning generation.
>
> Thank you for raising this point! This is correct for generalization across non-Wikipedia domains, and we will acknowledge this in the limitations section.
>
> However, for Wikipedia-domain generalization (Our vs. WikiIO), we compare task-specific vs. generic MGT within the same domain. This allows us to isolate the effect of the generation setup. From this comparison, we can conclude that task-specific Wikipedia data generalizes to generic Wikipedia data, but not the other way around.
>
> ---
>
> > Weakness 2: The benchmark, though covering crucial real-world tasks, omits others like translation and creative writing, and the scope of domains remains limited to Wikipedia.
>
> This is a clear limitation of our work, and we already acknowledge it in our limitations section.
>
> ---
>
> > Weakness 3: The paper misses an opportunity to provide examples of generic MGT cases and does not explore balancing task-specific and generic data for optimal detector training.
>
> We will follow your suggestion and have added MGT examples to the appendix. You are right that finding the right balance between generic and task-specific data is an interesting research question. We will add this to our future work section.
>
> ---
>
> > Weakness 4: The quality of the LLM-based translation is not checked.
>
> This is correct. We would like to clarify that we only use translation to evaluate factuality for prompt selection in Section 2.1.
>
> The lack of evaluating translated texts was also noted by reviwer gz51, which is why we added a translation quality check to the appendix (kindly see our response to Reviewer gz51).
>
> ---
>
> > Question 1: Some details are unclear, such as the specific model used for BERTScore, the references for per-language fine-tuned models, and the definition of "Wiki IO." in Exp. 3.
>
> Thank you! We have added these details to our manuscript.
>
> ---
>
> > Question 2: The prompt evaluation relies on a single model, limiting the reliability of its conclusions.
>
> This point was also raised by Reviewer TeVN. Kindly refer to our response above.
>
> ---
>
> > Question 3:  The reliability of the RAG pipeline, particularly the relevance of retrieved chunks, is not verified.
>
> That is correct. We do not perform an additional check because the prompt evaluation for RAG versus the baseline implicitly assesses whether the retrieved chunks improve text generation. As Table 1 shows, this is indeed the case: we observe higher scores across all evaluation dimensions, especially for factuality. This suggests that the retrieved chunks provide relevant context.
>
> ---
>
> > Question 4:  The justification for the specific language selection is currently poor and needs strengthening.
>
> This is a very valid point, thanks! Our choice is guided by the goal of including high-, medium-, and low-resource languages, thereby informing communities beyond the English Wikipedia about the potential and limitations of AI detectors. Concretely, we selected languages that vary along two key dimensions. Firstly, the number of active editors. Secondly, we aimed to choose languages for which the resource ranking on Wikipedia aligns with the resource ranking in common LLM training corpora. To verify this, we relied on language distribution statistics from Common Crawl (see [here](https://commoncrawl.github.io/cc-crawl-statistics/plots/languages)). The table below shows the metrics according to which we have chosen the languages:
>
> | Resource Level | Language   | Active Users | CC Share (%) |
> |----------------|------------|--------------|---------------|
> | High           | English    | 224,120      | 44.2668       |
> | Medium         | Portuguese | 8,343        | 2.1796        |
> | Low            | Vietnamese | 3,686        | 1.0326        |
>
> We have added this table and our motivation to our manuscript.
>
> ---
>
> > Question 5:  Below are some suggestions to fix typos and text errors: — L384 “accuracy of 89.7%”, in Appendix D.1 captions for figures are the same. — Appendix A: “DATA SET STATISTICS” –> “BENCHMARK STATISTICS”; “Corpus denotes the size of the data” –> “Corpus denotes the source of the data”. — Rephrase L070 “content integrity of UGC”. — Formulation “generic generation” is not clear. — The subsection "PROMPT EVALUATION" is missing a section number
>
> Thank you, we have incorporated these suggestions.

---

### Official Review · Reviewer_gz51 · 2025-10-30

**Soundness:** 2
**Presentation:** 2
**Contribution:** 3
**Rating:** 4
**Confidence:** 3

**Summary:**

This paper introduces TSM-Bench, a multilingual, multi-generator, and multi-task benchmark designed to evaluate machine-generated text (MGT) detectors in real-world Wikipedia editing scenarios. Unlike prior benchmarks that rely on generic prompts (e.g., "Write an article about X"), TSM-Bench focuses on task-specific LLM usage—such as paragraph writing, continuation, summarization, and text style transformation across English, Portuguese, and Vietnamese. The benchmark comprises approximately 152,000 texts and evaluated using 12 detectors. The authors find that existing detectors suffer a 10–40 percentage point drop in accuracy on task-specific MGT, exhibit asymmetric generalization (models trained on task-specific data generalize to generic data, but not vice versa), and tend to overfit to superficial artifacts when trained on generic outputs. Overall, the study compellingly argues that realistic, editor-aligned benchmarks are essential for assessing detector reliability on user-generated content (UGC) platforms like Wikipedia.

**Strengths:**

- The paper addresses a pressing concern: maintaining content integrity in Wikipedia and similar UGC ecosystems in the era of LLM-generated text. The motivation is clear and important for both the NLP and broader AI ethics communities.
- Comprehensive evaluation across 12 detectors and 5 experimental dimensions; inclusion of SHAP analysis gives interpretable evidence of overfitting to surface artifacts.
- The observed generalization asymmetry, where task-specific detectors generalize to generic data but not vice versa, is a notable and reproducible pattern that could inform future model training paradigms.

**Weaknesses:**

- The paper convincingly grounds its task design in empirical studies of Wikipedia editors'  LLM use and policy structures such as Manual of Style and Neutral Point of View. However, the benchmark ultimately treats detection as binary classification (HWT vs. MGT), despite acknowledging blended paragraphs where human- and machine-generated text co-exist. This abstraction may be practical for dataset construction, but undermines the ecological validity of the "real-world" claim. Real Wikipedia editing may involve mixed authorship, such as post-editing of LLM drafts, LLM-assisted revisions of human text, and stylistic corrections. A benchmark faithful to these workflows would ideally include segment-level or mixture-aware labeling to capture such dynamics. Thus, while the task types are empirically grounded, the evaluation protocol remains synthetically simplified.
- I kindly recommend that the authors improve the manuscript's presentation quality. For example, several key abbreviations and table entries are used without any definition at first mention, forcing readers to infer their meaning:
  - "TST" first appears in Section 2.2, where it evidently denotes "Text Style Transformation", but the acronym is not explicitly defined in the caption of Figure 6 and again in Section 5. This redundancy and delayed definition confuse the reader and suggest a lack of editorial consistency. Readers must deduce its meaning from context or later descriptions.
  - "English P." appears in Table 1 but is never defined. From the nearby mention of "paragraph-level English samples," one can infer that it stands for English Paragraph, yet this is not stated anywhere in the text or legend.
Such omissions may appear minor, but they significantly reduce readability for new readers and hinder reproducibility, as one cannot easily map metrics to task definitions.
- The factuality evaluation for Portuguese and Vietnamese data relies on GPT-4 translation followed by QAFactEval (line 212–213). However, the paper does not justify or quantify how this translation step affects factual consistency. Without back-translation checks or metric comparisons, it is unclear whether observed performance differences reflect model ability or translation artifacts.
- Prior work (such as M4GT, MAGE, and MULTITuDE) already explores multilingual or task-specific MGT; contribution is incremental without a stronger methodological leap.

**Questions:**

- Could the benchmark be extended to mixture-aware or segment-level labeling to better reflect real Wikipedia editing?
- How significant is translation bias in PT/VI QAFactEval via GPT-4?
- In what specific aspects does TSM-BENCH advance beyond prior work (such as M4GT, MAGE, and MULTITuDE) in design or analysis?
- Did supervised models trained on English task-specific data show any zero-shot gains on Portuguese or Vietnamese?
- Could the authors expand on whether overfitting features are artifacts of Wikipedia formatting specifically or general stylistic cues common across corpora?


If the authors can substantively address the questions and weaknesses outlined above, I would be inclined to raise my score.

---

> ### Author Response · Authors · 2025-11-20
> **Official Comment by Authors Part 1/2**
>
> We appreciate your review and thank you for recognizing the contribution of our work. Please find our responses to your feedback below:
>
> _We will soon upload an updated manuscript that incorporates the suggestions addressed in our rebuttal._
>
> ---
>
> > Weakness 1: Segment-level or Mixture-aware labeling
>
> This is a very relevant point, which also mirrors Reviewer qQWg’s comment. We fully agree that Wikipedia editing is nuanced and often involves mixed authorship or LLM-assisted revisions. This is precisely the motivation behind our four task formulations, which reflect different forms of human–LLM interaction.
>
> We decided to model detection for all four Wikipedia tasks using a binary framework for the following reasons:
>
> 1. **Practical detection objective**: Our benchmark is designed with the envisioned use of AI detectors on user-generated content platforms such as Wikipedia in mind. In such settings, the central decision is whether a revision unit should be flagged for potential AI authorship. While segment-level labels indeed offer deeper insight into the fine-grained dynamics of mixed authorship---as you rightly point out---the practical need for most platforms remains binary: flag or not flag. Therefore, we would argue that binary labeling aligns with the practical objective of supporting editors in identifying potentially machine-generated content.
> 2. **Mixed text as MGT**: We follow recent work that explicitly treats mixed-authorship text as machine-generated for evaluation and performs binary classification (e.g., Wu et al., 2025; Artemova et al., 2025; Saha & Feizi, 2025). Reviewer qQWg referred to LLM-as-a-Coauthor (Zhang et al., 2024) as an example suggesting binary classification is inadequate. In fact, Zhang et al. (2024) model both binary and three-class classification, and show that binary detection on mixed text achieves "profound" results (page 7).
>
> Overall, both prior work and our benchmark treat mixed-authorship text within a binary detection framework because the underlying goal is to determine whether a text is sufficiently machine-generated to warrant being flagged as MGT. This corresponds directly to the core objective of automated tools on user-generated content platforms, such as Wikipedia.
>
> Regarding Question 4: our benchmark can naturally be extended with mixture-aware or segment-level labels. We will explicitly mention this in the future-work section and plan to consider such extensions when expanding the benchmark to additional tasks where finer-grained annotations may yield further insights.
>
> ---
>
> > Weakness 2: Improving the manuscript's presentation quality
>
> Thanks! We have noticed further smaller naming inconsistency and have amended as you suggested.
>
> ---
>
> > Weakness 3: Evaluation of GPT-4 translation
>
> This is an excellent point, thank you.
>
> First, we would like to clarify that the detection of Portuguese and Vietnamese MGT is performed on the original machine-generated texts, not on their translations. The only use of translation occurs when identifying optimal prompts in Section 2.2. In that step, we evaluate whether task-specific prompts yield higher factuality compared to a generic baseline. Because most factuality metrics operate only in English, we translate the Portuguese and Vietnamese outputs solely for factuality evaluation of prompts, not for detection.
>
> The fact that (1) QAFactEval scores for Portuguese and Vietnamese are in the same range as for English, and (2) the magnitude of improvement over the baseline mirrors the English results (Table 1), suggests that translation bias is minimal.
>
> To further address your concern, we conducted an additional quality assessment using back-translation with the [DeepL API](https://www.deepl.com/en/pro-api), one of the leading commercial translation services. We include the results below and provide the full table in the appendix. High BLEU, ROUGE, and BERTScore values indicate that translation quality is sufficiently high and unlikely to affect prompt-selection outcomes. This is reinforced by the consistent factuality-prompt evaluation patterns observed across all three languages.
>
>
> | File            | BLEU      | ROUGE-1  | ROUGE-2  | BERTScore |
> |-----------------|-----------|----------|----------|-----------|
> | Paragraphs (PT) | 71.50     | 0.7838   | 0.6468   | 0.9671    |
> | Paragraphs (VI)   | 67.95     | 0.8475   | 0.7164   | 0.9486    |
> | Summarization (PT)  | 74.24     | 0.8767   | 0.8013   | 0.9736    |
> | Summarization (VI)     | 63.46     | 0.8868   | 0.7520   | 0.9411    |
>
> ---

---

> ### Author Response · Authors · 2025-11-20
> **Official Comment by Authors Part 2/2**
>
> > Weakness 4: Prior work already explores multilingual or task-specific MGT
>
> We appreciate the opportunity to clarify and sharpen how our benchmark differs from prior MGT detection datasets (especially the work you cited: M4GT, MAGE, MULTITuDE). Although we cite relevant prior work in the introduction, we agree that the distinction between our contributions and earlier benchmarks should be made more explicit.
>
> Regarding the three papers you mention: while they indeed explore multilingual settings, their methodological focus differs fundamentally from ours. None of the cited benchmarks incorporate task-specific generation. Instead, they employ precisely the type of generic prompt-based generation that we identify aim to overcome.
>
> The prior benchmarks rely on generic generation settings, as illustrated below:
>
> - **M4GT-Bench (Wang et al., 2024):**
>   - Extends M4 (Wang et al., 2023) by introducing new detection tasks, but crucially not new generation tasks. Their detection tasks include binary detection, multi-way detection, and text change point detection.
>   - The way M4GT-Bench expands the dataset follows exactly the generic generation paradigm we argue should be moved beyond. For example, their Wikipedia summary generation (Wang et al., 2024, p. 21) prompts a model to generate a Wikipedia summary for "`<title>`" without conditioning on the article contents. This corresponds directly to the generic text generation setup defined in Section 2.1 of our paper.
>
> - **MAGE (Li et al., 2024):** MAGE likewise employs a generic generation setting. For the news domain, their prompt is: *"Write a news article with the following headline: `<headline>`."* This is precisely the setup we reproduce in Experiment 3 to construct generic news instances.
>
> - **MULTITuDE (Macko et al., 2023):** This benchmark focuses on multilingual MGT detection, not task-specific settings. It considers only a generic news domain with prompts such as: *"Write a news article in {language_name} [...]."* Again, this aligns exactly with the generic generation paradigm we describe.
>
> Thank you again for this feedback. We have updated the manuscript to add a concise paragraph at the end of the related work section that explicitly delineates our methodological contribution relative to prior work.
>
> ---
>
> > Question 4: Did supervised models trained on English task-specific data show any zero-shot gains on Portuguese or
> Vietnamese?
>
> This is an excellent question and one we explicitly considered. We include three languages to enhance the generalizability and practical utility of our benchmark. However, we do not believe that our current language selection supports rigorous cross-lingual transfer evaluation at the depth and scope of specialised studies (e.g., Macko et al., 2023). For this reason, we focus on cross-task rather than cross-lingual experiments. We will include this in our limitations section.
>
> ---
>
> > Question 5: Could the authors expand on whether overfitting features are artifacts of Wikipedia formatting specifically or
> general stylistic cues common across corpora?
>
> This is another insightful question. We examined the MGT corpora across domains and languages and found that stylistic artifacts are rather corpus-specific. For example, in the news domain, one of the dominant artifacts is the use of bold styling cues. This aligns with the findings from Experiment 3. If supervised models were overfitting to the same cues across corpora, we would expect to see higher out-of-domain generalization in Experiment 3. However, the low cross-domain transfer indicates that the learned artifacts differ between corpora.
>
> ---
>
> > Question 1-3
>
> Please refer to our answers above.
>
> **References**
>
> Artemova, Ekaterina, et al. “Beemo: Benchmark of Expert-Edited Machine-Generated Outputs.” arXiv preprint arXiv:2411.04032, 2024.
>
> Doughman, Jad, et al. "Exploring the limitations of detecting machine-generated text." Proceedings of the 31st International Conference on Computational Linguistics. 2025.
>
> Lu, Ning, et al. "Large language models can be guided to evade ai-generated text detection." arXiv preprint arXiv:2305.10847 (2023).
>
> Pedrotti, Andrea, et al. "Stress-testing machine generated text detection: Shifting language models writing style to fool detectors." Findings of the Association for Computational Linguistics: ACL 2025. 2025.
>
> Saha, Shoumik, and Soheil Feizi. “Almost AI, Almost Human: The Challenge of Detecting AI-Polished Writing.” arXiv preprint arXiv:2502.15666, 2025.
>
> Wu, Junchao, et al. "Detectrl: Benchmarking llm-generated text detection in real-world scenarios." Advances in Neural Information Processing Systems 37 (2024): 100369-100401.
>
> Zhang, Qihui, et al. “LLM-as-a-Coauthor: Can Mixed Human-Written and Machine-Generated Text Be Detected?” arXiv preprint arXiv:2401.05952, 2024.

---

### Official Review · Reviewer_qQWg · 2025-11-01

**Soundness:** 2
**Presentation:** 2
**Contribution:** 2
**Rating:** 2
**Confidence:** 4

**Summary:**

This paper introduces TSM-BENCH, a multilingual, multi-generator, and multi-task benchmark for detecting machine-generated text (MGT) in realistic Wikipedia editing scenarios. The key insight is that existing benchmarks primarily evaluate detectors on generic text generation (e.g., "Write an article about X"), while real-world Wikipedia editors use LLMs for specific constrained tasks such as summarization, paragraph writing, and style transfer. The authors demonstrate that task-specific MGT more closely resembles human-written text (HWT) than generic MGT, making detection significantly more challenging.

**Strengths:**

1. The paper makes a valuable contribution by shifting focus from generic to task-specific MGT detection, which better reflects real-world LLM usage patterns. The identification of generalization asymmetry between task-specific and generic training data is an interesting finding. The multi-dimensional benchmark design (multilingual, multi-task, multi-generator) is comprehensive.
2. The paper is well-written with clear motivation and effective use of figures (especially Figure 1 and Figure 2). The experimental setup is described in detail, and results are presented in an organized manner. The distinction between generic and task-specific MGT is clearly articulated.

**Weaknesses:**

**1. Data Provenance and Contamination Concerns**

A fundamental assumption of this work is that the sampled Wikipedia texts are purely human-written. However, recent studies have documented that Wikipedia already contains LLM-generated content mixed with human contributions. For instance, [Detecting LLMs in the Wild: A Field Study of Latent Misuse](https://openreview.net/pdf?id=R2bp8xLLao) and [Wikipedia in LLM era](https://arxiv.org/abs/2503.02879) provide evidence of LLM usage in Wikipedia editing. The paper does not address this contamination issue or provide any verification methodology to ensure the "human-written" samples are actually purely human-written. This is a critical concern that potentially undermines the benchmark's validity. The authors should at minimum: (i) discuss this limitation explicitly, (ii) provide some analysis or filtering approach to mitigate contamination risk, or (iii) acknowledge the uncertainty in their ground truth labels.

**2. Missing Comparison with Highly Related Work**

The paper fails to discuss or compare with WETBench (cited as Quaremba et al., 2025 in the references), which appears to be a highly related concurrent work also focusing on detecting task-specific MGT on Wikipedia. This is a significant omission. The authors should provide a detailed comparison explaining: (i) how TSM-BENCH differs from WETBench, (ii) whether there is overlap in the data sources, and (iii) how the findings complement or contradict each other. Without this comparison, it's difficult to assess the novelty and positioning of this work. Moreover, like "Wikipedia in the Era of LLMs" and "Studying the Role of LLMs on Wikipedia" also discuss the LLM usage in Wikipedia while do not get any discussion and citation.

**3. Oversimplified Task Formulation for "Real-World" Claims**

While the paper claims to reflect "real-world Wikipedia editing practices," the task formulations are actually quite coarse-grained and unrealistic:

- **Paragraph-level detection is insufficient**: Modern Wikipedia editing with LLMs rarely involves writing entire paragraphs or complete continuations purely with AI. Real-world usage is much more fine-grained, involving sentence-level or even token-level human-AI collaboration. Recent work on collaborative writing has extensively studied this: [Coauthor (Clark et al., 2018)](https://arxiv.org/pdf/2201.06796) and [LLM-as-a-Coauthor](https://arxiv.org/abs/2401.05952) all demonstrate that human-AI co-writing happens at much finer granularities.

- **Binary classification may not be appropriate**: For the Paragraph Continuation task, the authors mention "blending HWT and MGT" but still frame it as binary classification at the paragraph level. If the continuation mixes human and machine text, shouldn't the detection task be formulated at a finer granularity? Labeling an entire human-machine hybrid paragraph as simply "MGT" seems oversimplified and may not align with practical needs. You can refer to LLM-as-a-Coauthor and Llm-detectaive for fine-grained HWT and MGT detection.

- **Impact on results interpretation**: This coarse-grained formulation might partially explain why white-box and black-box methods show such poor performance (<60% accuracy in Table 2). These methods rely on statistical patterns that may be diluted when human and machine text are mixed at the paragraph level. The strong performance of supervised methods (which can learn task-specific artifacts) compared to zero-shot methods supports this concern. Is the low performance due to the inherent difficulty of the task, or due to the mismatch between the evaluation setup and the actual detection problem?

**4. Language Selection Lacks Justification**

The choice of Portuguese and Vietnamese as the multilingual evaluation languages is not well motivated. More widely-studied and resource-rich languages like Chinese or French would seem like natural choices, given that: (i) they have substantial Wikipedia presence, (ii) there are more available detection tools and LLMs for these languages, and (iii) they would increase the benchmark's applicability to a broader research community. The paper should provide clear rationale for the specific language selection beyond data availability.

**5. Limited Discussion of Related Work on Human-AI Collaborative Writing**

The related work section misses an important body of literature on human-AI collaborative writing and mixed-authorship text. Works on co-writing, human-in-the-loop generation, and collaborative authorship have direct relevance to the "Paragraph Continuation" task and the real-world editing scenarios the paper aims to capture. This omission weakens the positioning of the work within the broader context of human-AI collaboration research.

**Questions:**

See weakness above.

---

> ### Author Response · Authors · 2025-11-20
> **Official Comment by Authors Part 1/2**
>
> Thank you for your time and effort providing such extensive and valuable feedback, which we greatly appreciate. Please find our responses to your comments below.
>
> _We will soon upload an updated manuscript that incorporates the suggestions addressed in our rebuttal._
>
> ---
>
> > Weakness 1: Data Provenance and Contamination Concerns
>
> This is a very valid concern. We use the data collected in Quaremba et al. (2025), who have taken several measures to mitigate contamination. These include:
>
> 1. Both mWNC and WikiPS were collected prior to the introduction of ChatGPT on 30 November 2022.
> 2. For mWNC, they excluded all edits made by bots, identified via edit comments.
> 3. For WikiPS, they applied the following quality controls:
>    (a) they only included full articles, excluding stubs via the `stub` tag, and
>    (b) they retained only text blocks that contain at least one reference, ensuring a minimum standard of textual quality.
>
> Taken together, these measures substantially reduce the likelihood that the datasets contain machine-translated content. While it is impossible to perfectly rule out the presence of AI-generated text in human-written samples, their data collection process was designed to minimize this risk as much as possible.
>
> ---
>
> > Weakness 2: Omission of relevant work
>
> We cited Quaremba et al. (2025) in Section 2.3 because we use their data for TSM-Bench. Their paper was published in a dataset track, and WETBench serves as an example of what can be built on top of WikiPS and mWNC. While our work builds on their datasets, it differs from theirs in several important ways:
>
> 1. Formalization of generation settings: We provide a clear formalization of generic vs. task-specific MGT generation (Section 2.1).
> 2. New task: We introduce the Paragraph Continuation task (Section 2.1).
> 3. Additional detector: We add BiScope as an additional context-aware detector.
> 4. More LLMs: We extend the generation setup by adding DeepSeek and GPT-4o.
> 5. Extensive experimental evaluation beyond their main results:
>      - We evaluate state-of-the-art off-the-shelf detectors on their data and compare performance to generic Wikipedia data. We show that these detectors significantly underperform on task-specific text.
>       - We examine generalization within and across domains and identify a generalization asymmetry (Section 4.3), which motivates a deeper analysis.
>       - We analyse feature importance of supervised detectors, showing that prior benchmarks inadvertently encourage overfitting to MGT artefacts.
>       - We also study cross-task generalization to determine whether task-specific data share common detectable features.
>       - Importantly, TSM-Bench enables us to draw critical conclusions about the reliability of detectors on UGC platforms such as Wikipedia, something that prior work could not meaningfully address. We also discuss these implications and how they related to other benchmarks.
>
> We acknowledge that the distinction between our work and Quaremba et al. (2025) is not sufficiently clear in the current manuscript. We appreciate you pointing this out and will add a dedicated paragraph in the related work section that clearly explains this distinction.
>
> Lastly, we are happy to include "Wikipedia in the Era of LLMs" in the new human-AI collaboration related work section.
> However, "Studying the Role of LLMs on Wikipedia" appears to be a research proposal.
>
> ---
>
> > Weakness 3: Oversimplified Task Formulation for "Real-World" Claims
>
> We greatly appreciate the time and effort you invested in providing such detailed and critical feedback. While we only partially agree with this specific set of comments, we address each point carefully below.
>
> 1. **"Task formulations are unrealistic"**
>
> Our motivation is to move beyond generic MGT and evaluate how detectors perform on *task-specific* instances. If our task formulations were unrealistic, this would indeed undermine the purpose of our benchmark.
>
> For this reason, we base our tasks on real-world evidence of how editors actually use LLMs in practice (Ford et al., 2023; Zhou et al., 2025). Ford et al. (2023) analyze Wikipedia discussion forums and interview editors about their LLM usage; Zhou et al. (2025) conduct interviews and report concrete LLM-supported editing workflows. Our task choices directly align with these findings:

---

> ### Author Response · Authors · 2025-11-20
> **Official Comment by Authors Part 2/2**
>
> - **Paragraph Writing / Summarisation**
>   $\rightarrow$ Zhou et al. (2025): editors report “creating articles” as a use case
>   $\rightarrow$ Ford et al. (2023): editors report “drafting article stubs” and “aiding with information synthesis”
>
> - **Text Style Transfer (NPOV corrections)**
>   $\rightarrow$ Ford et al. (2023): editors report “improving language” as a common use case
>
> These tasks are further corroborated by prior NLP work that seeks to support editors in *exactly* these workflows—for example, Ashkinaze et al. (2024) for NPOV corrections (our TST task), Liu et al. (2018) for summarisation, and a large body of work on machine-assisted content writing (e.g., Shao et al., 2024).
>
> 2. **"Paragraph-level detection is insufficient"**
>
> We agree that real-world human–AI collaboration often involves fine-grained editing. However, this does not preclude editing at *other* levels of granularity, such as sentence-, paragraph-, or article-level generation.
>
> As noted above, empirical studies on editor–LLM interactions document usage across multiple levels. The work you cite supports this:
>
> - **LLM-as-a-Coauthor (Zhang et al., 2024):**  Focuses on fine-grained targeted edits, but this does *not* imply that higher-level drafting (e.g., paragraph writing) is unrealistic.
>
> - **Huang et al. (2025)** (your citation under Weakness 2): Shows LLM modifications at the word-, sentence-, and—crucially—**paragraph-level**, which directly confirms the realism of our tasks.
>
> Our benchmark explicitly includes both granularity levels: paragraph-level detection (Paragraph Writing and Summarisation) and sentence-level detection (TST). We deliberately considered both.
>
> 3. **"Binary classification may not be appropriate"**
>
> This issue was also raised by reviewer gz51. Kindly refer to our detailed response in their review.
>
> 4. **"Impact on results interpretation"**
>
> We believe this comment pertains specifically to the mixed-text setting (if not, we are happy to clarify further).
>
>  1. We agree with your intuition that mixing human and machine text dilutes AI-specific signals, making zero-shot detectors struggle.
>  2. However, even in the Introductory Paragraph task---where no mixing occurs---zero-shot detectors already perform poorly. This provides strong evidence for our main claim: task-specific MGT is inherently harder to detect than generic MGT, independent of mixture effects. Mixed-text settings further amplify this difficulty, as expected based on the intuition above.
>  3. Our current setup does not allow us to determine whether zero-shot methods would perform differently under a three-class labeling scheme (human / machine / mixed). This is an interesting direction for future work but lies outside the scope of this study.
>
> ---
>
> > Weakness 4: Language Selection Lacks Justification
>
> We agree that the motivation for our language selection was not clearly communicated. This point was also raised by reviewer mKYw. Please find our response to this below.
>
> ---
>
> > Weakness 5: Limited Discussion of Related Work on Human-AI Collaborative Writing
>
> Thank you. We acknowledge this point and have added a third category of related work on “LLM-assisted Editing on Wikipedia” to our related work section. This will be included in the revised manuscript, which we will upload shortly.
>
> ---
>
> **References**
>
>
> Ashkinaze, Joshua, et al. "Seeing like an ai: How llms apply (and misapply) wikipedia neutrality norms." arXiv preprint arXiv:2407.04183 (2024).
>
> Liu, Peter J., et al. "Generating wikipedia by summarizing long sequences." arXiv preprint arXiv:1801.10198 (2018).
>
> Ford, Heather, Michael Davis, and Timothy Koskie. “Implications of ChatGPT for Knowledge Integrity on Wikipedia.” Meta-Wiki, July 2023, https://meta.wikimedia.org/wiki/Research:Implications_of_ChatGPT_for_knowledge_integrity_on_Wikipedia.
>
> Shao, Yijia, et al. "Assisting in writing wikipedia-like articles from scratch with large language models." Proceedings of the 2024 Conference of the North American Chapter of the Association for Computational Linguistics: Human Language Technologies (Volume 1: Long Papers). 2024.
>
> Zhang, Qihui, et al. “LLM-as-a-Coauthor: Can Mixed Human-Written and Machine-Generated Text Be Detected?” arXiv preprint arXiv:2401.05952, 2024.
>
> Zhou, Moyan, Soobin Cho, and Loren Terveen. "LLMs in Wikipedia: Investigating How LLMs Impact Participation in Knowledge Communities." arXiv preprint arXiv:2509.07819 (2025).

---

### Official Review · Reviewer_TeVN · 2025-11-02

**Soundness:** 3
**Presentation:** 3
**Contribution:** 3
**Rating:** 4
**Confidence:** 3

**Summary:**

The authors introduce TSM-BENCH, a new benchmark for evaluating machine-generated text (MGT) detectors in realistic Wikipedia editing scenarios. TSM-BENCH comprises 152,910 machine-generated texts across three languages, four editing tasks grounded in actual Wikipedia editing practices, and six different language models. Results show taht Off-the-shelf detectors achieving over 93% accuracy on generic MGT drop to between 47-73% on task-specific data. The authors discover a "generalization asymmetry": detectors trained on task-specific data can generalize to generic data and even across domains, but detectors trained on generic data fail catastrophically when applied to task-specific scenarios. Through feature analysis, the authors find that Models trained on generic data overfit to superficial artifacts like section formatting markers rather than learning genuine linguistic patterns that distinguish machine from human text.

**Strengths:**

1. TSM-BENCH is large-scale and comprehensive, featuring 152,910 machine-generated texts across 3 languages, 4 tasks, and 6 different generators.

2. The paper identifies a significant limitation in existing machine-generated text (MGT) detection benchmarks: previous work has primarily focused on generic text generation, while real-world users employ LLMs for specific, constrained tasks.

3. The authors evaluate 12 different detectors across multiple model families and conducts five comprehensive experiments including zero-shot performance, supervised learning, out-of-domain generalization, feature analysis, and cross-task transfer. The findings are significant and help explain why existing detectors may be unreliable in real-world settings.

**Weaknesses:**

1. While authors adapt prompts from the natural language generation literature shown to be most effective for each task, this might not reflect how people using LLMs for the task.

2. The study relies heavily on accuracy as the primary metric, with F1 scores provided but not deeply analyzed. In AI content detection task,  metrics like false positive rates also provide valuable insight.

3. The feature analysis in Experiment 4 shows that mDeBERTa trained on generic data overfits to surface-level features, while the model trained on task-specific data focuses more on semantically meaningful tokens. However, this analysis only examined mDeBERTa, Is this pattern of feature learning generalizable across other detection model families?

4. The prompt evaluation uses GPT-4o Mini as the sole judge for selecting the best prompts. This introduces potential biases and lacks validation against human judgments or alternative automatic metrics, which could affect the quality assessment of generated text.

**Questions:**

See weakness.

---

> ### Author Response · Authors · 2025-11-20
>
> Thank you for your overall positive assessment of our work and for your valuable feedback! Please find our responses to your comments below.
>
> _We will soon upload an updated manuscript that incorporates the suggestions addressed in our rebuttal._
>
> ---
>
> > Weakness 1: Prompts might not reflect how people use LLMs
>
> This is a very relevant concern. Indeed, our motivation would be undermined if the NLG prompts we use did not reflect how Wikipedia editors actually interact with LLMs.
>
> We would like to clarify that the prompts we use are basic in-context learning (ICL) prompts. We intentionally restricted ourselves to such simple prompts and did not employ more sophisticated techniques such as SICO (Lu et al., 2024), as used for example in Wu et al. (2025). While we do not have direct evidence that editors consistently use ICL prompts, their simplicity, popularity, and ease of use make them plausible in real-world editing scenarios.
>
> Finally, we found multiple examples on Wikipedia discussion pages where editors experiment with generating text using ICL-style prompts. For instance, see [this example](https://en.wikipedia.org/wiki/User:DraconicDark/ChatGPT).
>
> ---
>
> > Weakness 2: Other metrics would provide valuable insights
>
> This is absolutely correct. We focused on accuracy because our sample is balanced, and we applied this choice consistently throughout the paper. For readers interested in additional metrics, we also report F1-scores, although accuracy remained our primary measure.
>
> We fully agree with your suggestion. In response, we have added precision–recall curves to the appendix to provide deeper insight. These curves are presented for the best-performing model in our benchmark on GPT-4o-generated text, across all tasks and languages.
>
> ---
>
> > Weakness 3: Is the generalization asymmetry detectable across other detection model families?
>
> This is an excellent question, and we are actively investigating it in a separate research project. Recent work (Doughman et al., 2025; Pedrotti et al., 2025) has begun exploring the cues that detectors rely on. Motivated by the generalization asymmetry we identified, we are now working to understand what detectors actually learn. We believe this question is substantial enough to warrant a dedicated project.
>
> ---
>
> > Weakness 4: Prompt evaluation using only GPT-4o Mini lacks validation.
>
> We fully agree that without a thorough prompt evaluation, prompt choice can introduce bias and negatively affect the quality of generated text.
>
> To address this, we conduct a comprehensive evaluation of prompts across languages in Section 2.1. We assess three dimensions of generated text—n-gram overlap, factuality, and semantic similarity—using standard NLG metrics. Table 1 confirms a well-established finding in the prompting literature: ICL prompts tend to produce higher-quality text. Our prompt evaluation with GPT-4o-mini can therefore be seen as a (largely unsurprising) confirmation of this pattern in our task-specific generation setup.
>
> ---
>
> **References**
>
> Doughman, Jad, et al. "Exploring the limitations of detecting machine-generated text." Proceedings of the 31st International Conference on Computational Linguistics. 2025.
>
> Lu, Ning, et al. "Large language models can be guided to evade ai-generated text detection." arXiv preprint arXiv:2305.10847 (2023).
>
> Pedrotti, Andrea, et al. "Stress-testing machine generated text detection: Shifting language models writing style to fool detectors." Findings of the Association for Computational Linguistics: ACL 2025. 2025.
>
> Wu, Junchao, et al. "Detectrl: Benchmarking llm-generated text detection in real-world scenarios." Advances in Neural Information Processing Systems 37 (2024): 100369-100401.

---

### Author Response · Authors · 2025-12-02
**Summary of the rebuttal period**

Dear AC, dear reviewers TeVN, qQWg, gz51, and mKYw,

We thank all reviewers for their constructive feedback!

We appreciate that the reviewers acknowledged the contribution of our work, and we believe our rebuttal directly addresses the clarifications and concerns raised. Below, we summarise the received feedback and our responses. We note the low score from reviewer qQWg and emphasise that both concerns raised (missing related work and “oversimplified task formulation”) are thoroughly addressed in our rebuttal. In particular, for the latter point, we provide strong evidence refuting the claimed weakness, substantiated through multiple recent MGT detection benchmarks.

Overall, the requested additions mainly concern providing more insightful metrics (Reviewer TeVN), a clearer justification of our language selection (Reviewers qQWg and mKYw), and a sharper distinction from prior work (Reviewers gz51 and qQWg). We have incorporated this feedback and uploaded a revised manuscript. Specifically, we have revised the paper as follows:

1. **Translation quality**
   We conducted a back-translation quality assessment of all translated prompts (which we emphasise are used **only** for prompt-factuality evaluation; all detection experiments use the original source-language texts). We now cite Appendix Table 10 in Footnote 4, showing that translation bias is minimal.

2. **Language justification**
   We added our rationale for language selection to the introduction and included an appendix table (Table 5) reporting the metrics guiding our choices.

3. **Additional metrics beyond accuracy and F1**
   Following Reviewer TeVN’s suggestion, we added precision–recall curves for both supervised detectors across tasks and languages. We reference these additions at the end of the experimental-setup section (where we introduce the main evaluation metrics) and again at the beginning of Section 4.2.

4. **Related work**
   We added a new subsection on *AI-Assisted Editing on Wikipedia*. We clarified our use of the dataset from Quaremba et al. (2025) and sharpened the distinctions to prior work, especially M4GT, MAGE, and MULTITuDE, as suggested by Reviewer gz51. We also provide prompt examples from prior work in Appendix A.1 to better illustrate the reliance on generic setups.

5. **Minor manuscript improvements (e.g., naming)**
   We corrected naming inconsistencies and spelling issues throughout the manuscript.

We again thank the AC and reviewers for considering our work. We would like to stress that we believe that our rebuttal thoroughly addresses the weaknesses mentioned by the reviewers. Importantly, most requested additions (language selection, expanded related work, additional metrics) do not concern the core motivation or central findings of our work: that we must move beyond generic MGT detection, and that current detectors struggle in real-world scenarios.

Best wishes

The Authors

---

### Meta-Review · Area_Chair_qoXS · 2026-01-13

**Summary:**

The paper proposes TSM-Bench, a comprehensive benchmark designed to evaluate Machine-Generated Text (MGT) detectors within the context of realistic Wikipedia editing tasks (e.g., summarization, text style transfer, paragraph continuation). The authors demonstrate that state-of-the-art detectors, which perform well on generic text generation, suffer significant performance drops (10-40%) on these task-specific instances. A key contribution is the identification of a "generalization asymmetry": detectors trained on task-specific data generalize well to generic data, but the reverse is not true.

**Reviewer Concerns:**

Data Provenance and Contamination (Reviewer qQWg): This was the most critical concern raised by the dissenting reviewer, questioning whether the "human" text in the dataset was truly human-written given the proliferation of LLMs on Wikipedia.
Resolution: The authors clarified that they utilize the dataset from Quaremba et al. (2025), which employs strict filtering measures to mitigate contamination. These measures include using data collected prior to the release of ChatGPT (Nov 2022), excluding bot edits, and enforcing quality controls (e.g., reference requirements). The AC finds this explanation sufficient to uphold the validity of the ground truth.

Comparison to Related Work (Reviewers qQWg, gz51): Reviewers noted a lack of comparison to WETBench and other concurrent works.
Resolution: The authors clarified that TSM-Bench builds upon the data from WETBench but distinguishes itself through new task formulations (Paragraph Continuation), additional detectors (BiScope), more advanced LLMs (DeepSeek, GPT-4o), and a novel analysis of generalization asymmetry and feature importance.

Language Selection & Translation Quality (Reviewers qQWg, gz51, mKYw): Resolution: The authors justified their language choice (English, Portuguese, Vietnamese) based on resource availability (High/Medium/Low) and active user base. Furthermore, they conducted back-translation quality assessments (reporting high BLEU/ROUGE scores) to address concerns that translation artifacts might skew the factuality evaluation of prompts.

Metrics (Reviewer TeVN): The authors added Precision-Recall curves to supplement the accuracy metrics as requested.


Partially Addressed Concerns: Task Formulation (Binary vs. Mixed): Reviewers qQWg and gz51 argued that treating detection as a binary classification is an oversimplification, as real-world editing often involves mixed human-AI authorship. While the authors acknowledge this limitation, they argue that binary "flagging" remains the primary practical utility for platform maintainers. The AC agrees that while mixed-authorship detection is an important future direction, the current binary benchmark provides significant value in exposing the failure modes of current detectors on constrained generation tasks.

**Reviewer Scores:**

Reviewer mKYw (8): Likely remains 8 (Strong Accept).

Reviewer TeVN (4): Likely improves to 6 (Weak Accept).

Reviewer gz51 (4): Likely improves to 5 or 6 (Weak Accept).

Reviewer qQWg (2): Likely improves to 4 or 5.

---

### Decision · Program_Chairs · 2026-01-26

Accept (Poster)